# Model Approach to Thermal Conductivity in Hybrid Graphene–Polymer Nanocomposites

**DOI:** 10.3390/molecules28217343

**Published:** 2023-10-30

**Authors:** Andriy B. Nadtochiy, Alla M. Gorb, Borys M. Gorelov, Oleksiy I. Polovina, Oleg Korotchenkov, Viktor Schlosser

**Affiliations:** 1Faculty of Physics, Taras Shevchenko National University of Kyiv, 01601 Kyiv, Ukraine; nadtku@univ.kiev.ua (A.B.N.); alla.gorb@knu.ua (A.M.G.); fantality@ukr.net (O.I.P.); olegkorotchenkov@knu.ua (O.K.); 2Chuiko Institute of Surface Chemistry, NAS of Ukraine, 17 General Naumov Str., 03164 Kyiv, Ukraine; bgorel@ukr.net; 3Erwin Schrödinger International Institute for Mathematics and Physics, University of Vienna, 1090 Vienna, Austria; 4Department of Electronic Properties of Materials, Faculty of Physics, University of Vienna, 1090 Wien, Austria

**Keywords:** multilayer graphene, anatase, epoxy, polymer nanocomposites, thermal conductivity, Kapitza thermal boundary resistance

## Abstract

The thermal conductivity of epoxy nanocomposites filled with self-assembled hybrid nanoparticles composed of multilayered graphene nanoplatelets and anatase nanoparticles was described using an analytical model based on the effective medium approximation with a reasonable amount of input data. The proposed effective thickness approach allowed for the simplification of the thermal conductivity simulations in hybrid graphene@anatase TiO_2_ nanosheets by including the phenomenological thermal boundary resistance. The sensitivity of the modeled thermal conductivity to the geometrical and material parameters of filling particles and the host polymer matrix, filler’s mass concentration, self-assembling degree, and Kapitza thermal boundary resistances at emerging interfaces was numerically evaluated. A fair agreement of the calculated and measured room-temperature thermal conductivity was obtained.

## 1. Introduction

Polymer nanocomposites based on graphene and its derivatives remain the focus of numerous theoretical and experimental multidisciplinary types of research, which describe some novel structural [1,2,3,4], mechanical [5,6,7], electrical [8,9,10], and thermal [11,12,13,14] properties of the composites.

Such attention to the nanocomposites is fed by their promise as high-efficiency materials in a very wide range of applications. These include, for example, thermal interface materials [15,16,17], active electrodes for energy storage devices such as Li-ion batteries [18,19,20] and supercapacitors [21,22,23], membranes for fuel cells [24,25,26,27,28], gas sensors [29,30], and electromagnetic shields [31,32,33].

The state-of-the-art investigation of polymer nanocomposites has advanced over the past decade and generated several comprehensive review articles [34,35,36,37,38,39,40,41]. The key questions raised are how the filler content impacts the overall properties of the composite and to what extent the polymer–graphene interphase interaction variations, both with the morphology and surface activity of the filler, can be accounted for.

Recently, polymeric nanocomposites based on graphene/metal oxide hybrid nanostructures have been studied intensively as promising materials for the various above-mentioned applications [23,42,43,44,45,46,47,48,49]. A great variety of nano-sized metal oxides have been tested as anchoring layers on graphene and graphene oxide nanoplatelets, graphene-wrapped or graphene-encapsulated hybrid particles for fabricating advanced composite electrode materials for lithium-ion batteries [42,43], and electrochemical capacitors [23,42]. Examples of oxides include (but are not limited to) simple oxides (Cu_2_O, MnO, NiO, ZnO, CuO, SnO_2_, CeO_2_, RuO_2_, TiO_2_, MoO_2_, MnO_2_, Fe_2_O_3_, Fe_3_O_4_, Mn_3_O_4_, Co_3_O_4_, V_2_O_5_, and MoO_3_) and complex oxides (LiFePO_4_, Li_3_V_2_(PO_4_)_3_, and Li_4_Ti_5_O_12_).

Among the other oxides employed in hybrid graphene/metal oxide and graphene oxide/metal oxide fillers, a unique place pertains to two well-known allotropies of titanium dioxide (TiO_2_), namely rutile and anatase. Today, graphene@TiO_2_- and graphene oxide@TiO_2_-based polymer nanocomposites are promising materials that are not only used in energy storage applications but also in photocatalysis [44,45], gas sensing [46], solar cells [47], water splitting [48], and proton exchange membranes [49]. Great attention has been paid to graphene@TiO_2_ and graphene oxide@TiO_2_ hybrid nanoparticles because titanium dioxide nanoparticles demonstrate nontoxicity and good compatibility with organic solvents and polymers, and they also display good hydrophilicity and mechanical and thermal stabilities [50,51,52]. Also, the particular surface defect structure of titanium dioxide [53,54] endows it with highly reactive surface sites, which influence the interfacial charge and phonon transfer kinetics in composites with embedded TiO_2_ [55,56]. Following the comments of Wu et al. [42], the functions and synergistic effects of graphene/metal oxide hybrid nanoparticles embedded into a polymer matrix, which are related to the composite’s transport properties, can be briefly summarized as follows:

(1) Graphene nanoplatelets play the roles of 2D supports to uniformly anchor or disperse metal oxides with well-defined sizes, shapes, and crystallinity;

(2) Graphene nanoplatelets act as 2D conductive templates to build a 3D conductive porous network to improve the poor electrical properties and charge transfer pathways of pure oxides;

(3) Graphene nanoplatelets suppress the volume change and agglomeration of metal oxide nanoparticles;

(4) Metal oxide nanoparticles suppress the re-stacking of individual graphene nanoplatelets;

(5) Metal oxide nanoparticles can promote both charge and phonon out-of-plane transfer in multilayered graphene (MLG) nanoplatelets by binding dangling edge carbon bonds of neighboring monoatomic layers with the active sites located on the metal oxide nanoparticles.

Typically, the synthesis of MLG-TiO_2_ nanocomposites includes in situ growth and covalent and non-covalent grafting [57]. One should bear in mind that the rational design of hybrid nanoparticle architecture and the control of the morphology and phase composition of metal oxides on graphene can ensure reproducibility and a better understanding of the structure–property relationships in the nanocomposites. Also, a clear perception of the surface chemistry on graphene and metal oxides is crucial for the enhancement of the interfacial synergistic effects that are favorable for improved phonon transport.

The thermal conductivity of the composites is restricted by the high thermal interfacial resistance originating from strong interfacial phonon scattering. To obtain a thermal interface material with nanocomposites containing high thermal conductivity fillers, their concentration would thus be rather high. However, as the material is expected to be electrically insulating to prevent the malfunction of electronic devices at high working voltages, this requires low filler concentrations. Most frequently, reasonable thermal conductivities in the nanocomposites will inevitably result in enhanced electrical conductivities due to a low percolation threshold. Meanwhile, an increased thermal conductivity with filler loading and simultaneously decreased electrical conductivity were reported to arise from the attachment of functionalized molecules or nanoparticles to the filler surface [58,59].

Assuming that the temperature field obeys a diffusive equation, the kinetic theory predicts the Fourier law to be q=−κΔT with the heat flux q, thermal conductivity κ, and temperature gradient ΔT. In solids, heat is carried by acoustic phonons and electrons so that the thermal conductivity can be represented as a two-term sum, $\kappa =\kappa_{ph}+\kappa_{e}$, with phonon ($\kappa_{ph}$) and electron ($\kappa_{e}$) contributions. In metals, $\kappa_{e}$ dominates due to a large concentration of free carriers (κ~380–400 W/m K in copper). In insulating crystals, heat is transported via lattice vibrations, and the Boltzmann–Peierls approach [60] became one of the cornerstones in the theory of lattice thermal conductivity $\kappa_{ph}$.

It is, however, generally agreed that heat conduction in carbon materials is usually dominated by phonons, even in graphite, which has metal-like properties [61]. This is readily explained by the strong covalent ${sp}^{2}$ bonding and resulting efficient heat transfer via lattice vibrations, an assertion for which there is prior evidence. However, it is likely that the contribution of $\kappa_{e}$ will be significant for the doped materials [62].

In the relaxation time (τ) approximation, various scattering mechanisms, which limit the phonon mean free path (Λ=τv, with v being the phonon group velocity, which is approximated by the sound velocity when the acoustic dispersion can be ignored), are additive for the i-th scattering processes, so that $\tau^{-1}=\sum_{i}\tau_{i}^{-1}$. The acoustic phonons are scattered by other phonons, lattice defects, impurities, free electrons, interfaces, etc.

The most elementary picture of heat conductivity is based on the analogy with the kinetic theory of gases when $\kappa_{ph}=\frac{1}{3}C_{p}v\Lambda$, where $C_{p}$ is the specific heat capacity. In a lattice, the oscillating particles are replaced with normal vibration modes, which possess well-defined wavelengths and frequencies. It is also necessary to take into account that the different modes obey different dispersion relations and thus have different group velocities $v_{j}$ depending on their wavenumber (or frequency). One can thus equate $\kappa_{ph}=\sum_{j}C_{j}\left(\omega \right)v_{j}^{2}\left(\omega \right)\tau_{j}\left(\omega \right)d\omega$, where $C_{j}$ is the heat capacity, and ω is the phonon frequency, and the summation extends over all j phonon polarization branches (two transverse and one longitudinal acoustic branch).

Thermal conductivity is called intrinsic when it is limited by the crystal lattice anharmonicity. The intrinsic κ limit is reached when the crystal lattice is perfect, and phonons can only be scattered by other phonons. The anharmonic phonon interactions, which lead to finite κ in three dimensions, can be described by the Umklapp processes. The degree of the anharmonic interatomic forces is characterized by the Grüneisen parameter γ, which enters the expressions for the Umklapp scattering rates. Thermal conductivity is termed extrinsic when it is mostly limited by extrinsic effects, such as phonon scattering on rough boundaries or lattice defects [62]. In epoxy–carbon composites, the electronic thermal conductivity is less than 2% of the thermal conductivity of the material and, therefore, may be considered negligible [63].

In nanostructures, κ is reduced by scattering from boundaries. Specular scattering with momentum conservation does not add to thermal resistance. Only diffuse phonon scattering from rough interfaces without momentum conservation (p → 0) limits the free path Λ.

If the boundary scattering dominates, $\kappa_{ph}\sim C_{p}v\Lambda \sim {C_{p}v^{2}\tau }_{B}\sim C_{p}vD$. At D≪Λ, phonon dispersion is modified due to confinement, resulting in a changing υ value and complicating the size dependence of $\kappa_{ph}$ [62]. $C_{p}$ is determined by the phonon densities of states, which are effectively modified by reducing the dimensionality. Consequently, $\kappa_{ph}\sim T^{3}$ is obtained at low temperatures in bulk materials and $\kappa_{ph}\sim T^{2}$ is seen in 2D systems.

It is unlikely, however, that the above equations can be easily generalized to the case of heterogeneous composites, such as polymer–graphene composites. Despite the long-standing interest in the development of a suitable model describing the heat transfer through the matrix–filler interphase, models of the thermal conductivity in such systems remains highly controversial [59]. Our understanding of the heat transport phenomena is even more rudimentary for functionalized fillers such as graphene/metal oxide fillers.

Here, we prepared epoxy nanocomposites with MLG and embedded nanoparticles of TiO_2_ and studied a temperature dependence of the thermal conductivity of the material in the range from about 40 to 300 K. The thermal conductivity of the composite was numerically computed as a function of the two-layer filler loading. The proposed soft-sphere approach and related concept of the effective thickness allowed us to simplify the thermal conductivity simulations by replacing a dotty interface with a continuous one and by describing interface thermal transport in the hybrid graphene@anatase TiO_2_ nanosheets via the phenomenological thermal boundary resistance.

## 2. Results and Discussion

### 2.1. Temperature Dependence of the Thermal Conductivity

Figure 1a shows the measured temperature dependencies of the thermal conductivity of the neat epoxy (open circles) and its composites filled with hybrid MLG@TiO_2_ nanoparticles with different filler contents, $C_{f1}:C_{f2}=$ 0.01:0.01 (closed circles) and 0.01:0.05 (squares). The experimental data show aggregate behaviors that are typical of the thermal conductivity in disordered materials. Indeed, we previously reviewed extensive studies on amorphous oxides [64].

As an example, the computed $\kappa \left(T\right)$ behavior in amorphous SiO_2_ (a-SiO_2_) is depicted in Figure 2. The $\kappa \left(T\right)$ dependence can be described by calculating the total mean free path Λ as a combination of various phonon scattering mechanisms using the Matthiessen rule,
(1)$$\frac{1}{\Lambda }=\sum_{i}\frac{1}{\Lambda_{i}},$$
where the sum runs over all scattering mechanisms, e.g., phonon–phonon, phonon–boundary, phonon–point defect scattering, etc.

In disordered materials, the glass-like $\kappa \left(T\right)$ dependence is characterized by the initial $\kappa {\sim T}^{2}$ rise at very low temperatures, followed by a characteristic plateau around 10 K (marked by arrow S in Figure 2) and a further slow increase, which is weaker than $\kappa {\sim T}^{2}$, with the increasing temperature [66]. Such $\kappa \left(T\right)$ dependence is exemplified in Figure 2.

The physical mechanisms that may be responsible for such behavior have been extensively debated. The $T^{2}$ dependence at the lowest T has been shown to be well described by the scattering on statistically distributed localized tunneling entities [67,68]. The plateau and gradually increasing κ at higher temperatures are well described by incorporating the soft-potential model [69,70].

The gradual increase in the thermal conductivity shown by the circles in Figure 1 is hence in qualitative agreement with the above prediction. The $\kappa \left(T\right)$ dependence given by the squares is observed to saturate above ≈ 150 K. A small plateau can also be seen between 150 and 200 K for the net epoxy (open circles). The inclusion of MLG@TiO_2_ nanoparticles increases the thermal conductivity (closed circles in Figure 1). Increasing the concentration of TiO_2_ shows a further enhancement in κ (squares compared with closed circles in Figure 1). It is not immediately obvious how the above models can be combined for the interpretation of the experimental results.

### 2.2. Development of the Model Formalism

In constructing a suitable model to describe the experimental data, we begin by picking the thermal conductivities of epoxy, graphene, and anatase TiO_2_ along with the thermal boundary resistances, which are given in Table 1.

It should be noted that the theoretical calculations based on Maxwell Garnett’s effective media approximation [75] showed that, at low loadings ($\varphi_{f1}\leq$ 2%), the thermal conductivity does not depend on the graphene–epoxy thermal boundary resistance ($r_{1}$) when it varies in the range from 0.35 × 10^−8^ $\leq r_{1}\leq$ 8.3 × 10^−8^ m^2^·K·W^−1^ (see Figure 4b in ref. [76]).

It has been widely observed that the introduction of the Kapitza thermal boundary resistance between the MLG nanoplatelets and a host polymer matrix into the model can provide readable accounts of the thermal conductivity experiments and theoretical approaches. In hindsight, the heat transfer enhancement in nanocomposites is largely determined by the thermal boundary resistance, but the conventional heat conduction models for the interfaces fail to have the correct value of the thermal boundary resistance.

We begin with the development of our model by introducing the Kapitza resistance [79] $R_{K}$ between the thermally conductive filler nanoparticle with the thermal conductivity $\kappa_{f}$ and the 3D polymer matrix ($\kappa_{m}$), which arises from the large difference in the phonon density of states of the low-dimensional and bulk media. The Kapitza resistance is in series with the particle thermal resistance $l_{f}/\kappa_{f}$, where $l_{f}$ is the filler particle size [77]. For simplicity, let us consider the equivalent thermal resistance as ${\left(\frac{l_{f}}{\kappa_{f}}\right)}_{eff}=\frac{l_{f}}{\kappa_{f}}+R_{K}$ and assume that the effective thermal conductivity of the particle is [77]
(2)$$\kappa_{f,eff}=\;\frac{\kappa_{f}}{1+\;\frac{R_{K}\kappa_{f}}{l_{f}}}\;=\;\frac{\kappa_{f}}{1+\;\frac{\eta_{f}}{l_{f}}}\;,$$
where $\eta_{f}$ is the Kapitza length. This can be introduced to describe the length scale, which is defined by the equality of the interface thermal resistance to the heat flux and the thermal resistance originating from the bulk [80]. The existence of the thermal boundary resistance leads to a temperature drop ∆θ across the interface and [80]
(3)$$R_{K}=\;\frac{∆\theta }{Q_{i}}\;=\;\frac{\eta_{m,f}}{\kappa_{m,f}}\;,$$
where $Q_{i}$ is the heat flux across the interface, whereas the subscripts m and f indicate the matrix and filler, respectively. The most widely accepted case for removing $R_{K}$ from the thermal conductivity expression is extending the problem to a large-scale system, when the interface thermal resistance can usually be neglected. By increasing the interparticle separation distance and filler size $l_{f}$ a smaller portion of ${\left(\frac{l_{f}}{\kappa_{f}}\right)}_{eff}$ is related to the interface.

Here, we approximate hybrid MLG@TiO_2_ nanoparticles using the two bounding surfaces denoted by the subscripts 1 (MLG) and 2 (TiO_2_) so that
(4)$$\frac{l_{f1}+l_{f2}}{\kappa_{h,eff}}=\frac{l_{f1}}{\kappa_{f1}}+R_{K1}+\frac{l_{f2}}{\kappa_{f2}}+R_{K2}=\frac{l_{f1}\kappa_{f2}+l_{f2}\kappa_{f1}+\left(R_{K1}+R_{K2}\right)\kappa_{f1}\kappa_{f2}}{\kappa_{f1}\kappa_{f2}}\;,$$
where $R_{K1}$ and $R_{K2}$ stand for the MLG-TiO_2_ and TiO_2_-epoxy interfaces, respectively. The effective thermal conductivity of the hybrid nanoparticle ($\kappa_{h,eff}$) is then readily obtained as
(5)$$\kappa_{h,eff}=\;\frac{\kappa_{f1}}{\frac{l_{f1}}{l_{f1}+l_{f2}}\cdot \left(1+\frac{\eta_{f1}}{l_{f1}}\right)+\frac{l_{f2}}{l_{f1}+l_{f2}}\cdot \left(1+\frac{\eta_{f2}}{l_{f2}}\right)\cdot \frac{\kappa_{f1}}{\kappa_{f2}}}\;.$$

However, the above approach implies the existence of a distinct boundary between the constituent phases. Moreover, there are no contributions in Equations (2)–(5) to account for the geometrical shape of the filling particles. Therefore, to account for the interphase layers surrounding the fillers of different shapes, the thermal conductivity of the interphase regions ($\kappa_{int}$) was determined using the Kapitza thermal boundary resistance $R_{K}$ as [12]
(6)$$\kappa_{int}=\;\frac{1}{h_{int}}\int_{0}^{h_{int}}\frac{h\cdot dh}{R_{K}}=\;\frac{h_{int}}{2R_{K}}\;,$$
where $h_{int}$ is the interphase layer thickness. It should be noted that, since the interphase layers are composed of segments of macromolecular chains, the values of $\kappa_{int}$ are supposed to be close to $\kappa_{m}$. It is therefore possible to roughly estimate a spread of the $h_{int}$ values provided that the corresponding $R_{K}$ value is known (or vice versa).

Using Equation (6), the above $\kappa_{f,eff}$ can be modified (see Appendix A). A graphical data analysis of both $\kappa_{f,eff}$ and $\kappa_{h,eff}$ at varying interphase layer parameters is performed in Section 2.5. The numerical estimates of $\eta_{f}$ and the normalized effective thermal conductivities $\kappa_{f,eff}/\kappa_{f}$ are given in Table 2 for the anatase, the in-plane ($\kappa_{g11}=\kappa_{g22}$) and cross-plane ($\kappa_{g33}$) thermal conductivities of the MLGs, and the hybrid particles of the sandwich-like MLG@TiO_2_ and of the graphene-wrapped anatase architectures. In the evaluation of $\kappa_{int}$, the value of 10 nm was taken for $h_{int}$ (see comments in Section 2.5).

### 2.3. Discussing the Temperature Dependence of the Thermal Conductivity in Nanocomposites

We fitted the measured $\kappa \left(T\right)$ data to a power-law dependence $\kappa \left(T\right)\propto T^{n}$. The applied analysis yields several temperature regions of such dependence with the marked exponent values of n, as given by the lines in Figure 1b.

For the epoxy resins, a linear temperature dependence $\kappa \left(T\right)\propto T$ was observed in the region from 10 to 300 K [81,82]. In our neat epoxy, n varies from about 0.4 to 0.5 upon increasing T (open circles in Figure 1b). This discrepancy can arise from a much lower cross-linking degree in our epoxy compared to that reported in the earlier studies.

Forming the epoxy/MLG@TiO_2_ composite does not affect the n values remarkably (closed circles and squares in Figure 1b). Meanwhile, increasing the TiO_2_ content exhibits the saturation of κ, which shows up prominently above T≈ 150 K (squares).

These tendencies may be compared with theoretical and experimental results from the literature. In particular, single-layer graphene nanoribbons follow the $\kappa \left(T\right)\propto T^{n}$ behavior below 100 K with the n value varying from 1.5 to 2.0 for different phonon modes [83,84]. Graphite behaves as $\kappa \left(T\right)\propto T^{2.5}$ [85]. The theoretical calculations for the anatase TiO_2_ single crystals show that $\kappa \left(T\right)\propto T^{-1}$ is within the temperature range of 100–500 K [86] with the dominating Umklapp phonon scattering. For the nanosized anatase particles, about 100 nm in diameter, the measured temperature dependence of κ exhibits a sinusoidal-like growth below 120 K, followed by a plateau in the range of 120–180 K with κ≈ 0.8 W m^−1^ K^−1^, and then it shows a gradual decrease [87].

It can therefore be assumed from the above analysis that the behavior of κ in anatase TiO_2_ may have an impact on the saturation observed in the curve with the squares in Figure 1. As the $\kappa_{C}\left(T\right)$ dependence of our epoxy/MLG@TiO_2_ composite samples in the whole studied temperature range cannot be explained quantitatively by the temperature behavior of its constituents, the contributions of the filler–polymer interphase regions should then be taken into account. A substantial difference between the open and closed circle data in Figure 1 above ≈ 150 K can now be thought to be assigned to the increasing volume of the interphase region with the increasing mass concentrations of the MLG and TiO_2_ fillers.

More specifically, to calculate the temperature dependences of $\kappa_{f,eff}\left(T\right)$ (see Section 2.2) and to extract the partial contributions of MLG and TiO_2_ into Equation (2), the temperature dependence of the Kapitza resistance $R_{K}\left(T\right)$ for all interfaces in the composite is therefore needed to provide basic information for the understanding of the measured $\kappa_{C}\left(T\right)$.

There are two widely used models for evaluating $R_{K}$, the acoustic mismatch model (AMM) [88,89] and the diffuse mismatch model (DMM) [85,88,89]. Both models give $R_{K}\left(T\right)\propto T^{3}$ below 50 K if the phonon dispersion can be neglected [88]. However, the temperature dependence of the acoustic wave velocities incorporated into the models cannot be neglected at an increasing T, even if the velocity dispersion is neglected. Moreover, in the case of graphene, the phonon dispersion of the TA modes, which provide a dominant contribution to $R_{K}$, cannot be neglected either. Therefore, it may be expected that the $R_{K}\left(T\right)$ dependence may diverge from the $T^{3}$ trend in the temperature range of interest. It has been observed that $R_{K}\left(T\right)$ at the solid–solid interface decreases with an increasing T from 100 to 300 K, with a smaller decreasing rate above 200 K [90]. A decreasing $R_{K}$ value increases the thermal conductivity of the interphase region.

We may thus assume that a similar temperature behavior of $R_{K}$ is observed at the graphene–anatase ($R_{K}\equiv r_{2,1}$), graphene–epoxy ($R_{K}\equiv r_{1}$), and anatase–epoxy ($R_{K}\equiv r_{2}$) interfaces. Therefore, the increasing $\kappa_{C}$ with the increasing T, followed by their saturations at higher loadings, is due not only to the increasing $\kappa_{1,2}$ value but also to the decreasing (stabilizing) $r_{1,2}$. It may be suggested that this methodology can be used to describe $r_{1}(T)$ at the graphene–epoxy interface and then to incorporate $r_{1}(T)$, $r_{2}(T)$, and $r_{2,1}(T)$ in our model. However, a thorough theoretical analysis of $\kappa_{C}\left(T\right)$ is beyond the scope of this study.

### 2.4. Theoretical Simulation and Numerical Calculations

A great number of theoretical models for evaluating the thermal conductivity of composite materials have been developed [12]. To construct an appropriate model aimed to quantitatively analyze the temperature-loading dependences of κ in our nanocomposites, three assumptions were taken into account:

(1) The polymer matrix contains three solid phases, i.e., MLGs (phase 1), anatase TiO_2_ nanoparticles (phase 2), and self-assembled hybrid MLG@TiO_2_ nanoparticles (phase 3);

(2) Every particle embedded into the matrix is surrounded by an interphase layer, which has a molecular structure that is determined by the particle–matrix interaction and differs from the polymer structure beyond the layer;

(3) All phases are dispersed randomly and uniformly within the matrix.

We start from the model developed by Su et al. [12], where the authors obtained an algebraic equation to calculate the loading dependence of the thermal conductivity for graphene-filled polymer nanocomposites, taking the thermal boundary resistance into account. To adapt the original model of Su et al. to describe a nanocomposite filled with two-layer (“hybrid”) nanoparticles, a certain geometrical configuration of the hybrid particles should be assumed. For the case of graphene-based hybrid nanosheets, a large number of possible configurations have been proposed in the literature (for a brief survey, see, for example, [42]). Here, we use the so-called sandwich-like configuration (SLG) for the MLG@TiO_2_ hybrid nanosheets proposed earlier in ref. [91]. Different geometrical versions of the SLG, including parallelepiped-shaped or spheroid-shaped graphene nanoplatelets assembled with either hard or soft spherical anatase nanoparticles, are given in Figure 3 (see also Appendix A).

In our case of a three-phase composite material, the basic equation of the model (see Equation (4) in ref. [12]) takes the following form:(7)$$\sum_{n=0}^{3}\varphi_{n}(\rho_{n},{p_{1},p_{2},C}_{1},C_{2})\cdot \Phi_{n}(\left\{L_{n}\right\},C_{1},C_{2},r_{n},\kappa_{n},\kappa_{c})=\;0$$
where $\varphi_{n}$ represents the volume concentrations of the constituent phases (the subscripts n= 0, 1, 2, and 3 correspond to the polymer matrix, free MLG nanoplatelets, free anatase nanoparticles, and hybrid MLG@anatase nanoparticles, respectively) and
$$\sum_{n=0}^{3}\varphi_{n}=\;1.$$$C_{1}$ and $C_{2}$ are the mass concentration of graphene nanoplatelets and anatase nanoparticles, respectively, $p_{1}$ and $p_{2}$ are the relative mass portions of the free (unassembled) MLG and anatase particles, respectively, ($0\leq p_{1,2}\leq 1$), $\rho_{n}$ denotes the mass densities, $\left\{L_{1,2}\right\}$ is the set of geometrical sizes, $r_{1,2}\;\mathrm{and}\;r_{21}$ are the Kapitza thermal boundary resistances, $\kappa_{n}$ denotes the thermal conductivities ($\kappa_{1}$ includes both in-plane ($\kappa_{g11}$) and out-of-plane ($\kappa_{g33}$) components), and $\kappa_{c}$ is the thermal conductivity of the composite.

The parameters $p_{1}$ and $p_{2}$ have been introduced to describe the self-assembling effect itself. A case of $p_{1}=p_{2}=$ 1 means that the effect is absent (i.e., the case of two filling phases “1” and “2” in a host matrix), whereas the opposite extreme case of $p_{1}=p_{2}=$ 0 implies complete self-assembling when there is a single (hybrid) constituent phase “1@2” in a matrix. The intermediate situation of $0<p_{1,2}<1$ corresponds to a three-phase composite comprising fillers “1” and “2” and their hybrid “1@2”. It should be noted, however, that there is no direct evidence for the self-assembling of graphene nanoplatelets and anatase nanoparticles. This will require further experimental evidence to secure the validity of utilizing the self-assembling properties partly because of the fact that the non-oxidized graphene can only exist in a suspension.

The explicit expressions for the functions $\Phi_{n}$ and $\varphi_{n}$ are given in Appendix A, respectively. The functions have no obvious physical meaning and are used only to reduce Equation (7) to the compact form. In general, the model incorporates a whole number of both geometrical and material parameters related to the matrix and each constituent phase. The parameters are taken as independent variables to calculate the thermal conductivity $\kappa_{C}$ of the composite.

By using the model in the “soft-spheres” approximation, first of all, we complete a fitting procedure to match the calculated values of $\kappa_{C}$ to the measured values $\kappa_{C,E}$, supposing that an extreme case of complete self-assembling (i.e., $p_{1}=p_{2}=$ 0) takes place. In doing so, we input $\kappa_{m}$, $\kappa_{2}$, $\kappa_{g33}$, and $r_{1}$ from Table 1, and then assign fixed values for the interface layers’ thicknesses $h_{1}=h_{2}=$ 24 nm, the nanoplatelet’s aspect ratio α = 0.01 $\kappa_{g11},r_{1}$ and $r_{2}$, and then try to satisfy the target condition in the form $\kappa_{C}=\kappa_{C,E}$ by varying the graphene–anatase thermal boundary resistance $r_{21}$. The parameters used in the fitting procedure are summarized in Table 3.

As expected from Table 3, the resulting values of $r_{21}$ do not depend on $r_{1}$. Since the numerical values for $r_{2}$ are lacking in the literature, they were varied by up to one order of magnitude (the values in the brackets and without the brackets in Table 3). Decreasing $r_{2}$ increases $r_{21}$ from 11% to 12.6% for $C_{1}:C_{2}=$ 0.01:0.01 or from 46% to 52% for $C_{1}:C_{2}=$ 0.01:0.05, where the smaller percentages correspond to lower $\kappa_{g11}$ values. One can also see that the certain increment in $r_{21}$ is achieved while $\kappa_{g11}$ is doubled from 600 to 1200 W·m^−1^·K^−1^. The resulting magnitude of $r_{21}$ (from about 0.25 × 10^−7^ to 0.62 × 10^−7^ m^2^∙K^−1^∙W^−1^) seems to be appropriate. We can, for example, compare this value with the calculated room-temperature thermal boundary resistance of the Cu/Si interface, which is 5 × 10^−4^ m^2^∙K^−1^∙W^−1^ [90].

Experimentally, it is rather difficult to distinguish between the thermal resistance of a thin layer and the thermal resistance of the substrate. Zheng et al. [92] treated the graphite/Ni interfaces as one effective layer and fit the time domain thermoreflectance (TDTR) data to two free parameters. A total interface thermal resistance of ≈2.3 × 10^−8^ m^2^∙K^−1^∙W^−1^ was obtained. The effective cross-plane thermal conductivity of ≈3.3 W m^−1^ K^−1^ was reported, whereas the in-plane thermal conductivity varied between ≈650 and 1000 W m^−1^ K^−1^.

Multilayered graphene nanoplatelets typically contain structural defects, including point and edge defects, surface wrinkles, mesovoids, etc. The effects of defects on the interface’s thermal conductivity in graphene/epoxy composites were observed by employing molecular dynamics simulations [93]. Among the various defect types, Stone–Wales defects are the most effective in improving the interface’s thermal conductivity in the composite. This is due to the enhanced vibration intensity of the out-of-plane low-frequency phonons, which could enhance the phonon coupling between the epoxy and graphene, thus increasing the conductivity. In turn, increasing the number of graphene layers in the MLG enhances the phonon scattering, thus decreasing the thermal conductivity. When the layer number exceeds four, the MLG properties become close to that of graphite, and the interface thermal conductivity stabilizes at the value close to that of graphite.

The variation in the interfacial thermal resistance with respect to the functionalization, isotope, and acetylenic linkages in graphene is also systematically examined using molecular dynamics [94]. The graphene–epoxy interfacial thermal resistance can be remarkably reduced by using the covalent and noncovalent functionalization. Among different functional groups, covalent butyl is the most effective substance in reducing the resistance.

Strictly speaking, the thermal boundary resistance for the in-plane graphene–epoxy interface ($r_{1X}=r_{1Y}$) differs from the cross-plane resistance ($r_{1Z}$) (see Appendix A). These should be completely different thermal contacts since the graphene phonon density of states is shifted to a higher energy due to the in-plane vibration compared to the cross-plane case. The sound velocity is also significantly different in the in-plane and cross-plane configurations. Meanwhile, an average value $r_{1av}$ of $r_{1}$ can only be measured because the partial contributions of $r_{1X}$ and $r_{1Z}$ cannot be separated from each other. To obtain a quantitative estimate of the error incurred on the approximation of $r_{1}\approx r_{1X}=r_{1Z}$, the value of $r_{1av}$ for parallelepiped-shaped graphene nanoplatelets can be expressed as
(8)$$\frac{1}{r_{1av}}=\frac{\frac{L_{X}L_{Z}}{r_{1X}}+\frac{L_{Y}L_{Z}}{r_{1Y}}+\frac{L_{X}L_{Y}}{r_{1Z}}}{L_{X}L_{Z}+L_{Y}L_{Z}+L_{X}L_{Y}}.$$

Inputting $r_{1X}=r_{1Y}$, $L_{Z}=$ 50 nm, $L_{X}=L_{Y}=$ 5 × 10^3^ nm (average linear dimensions of our nanoplatelets) yields $r_{1}\approx$ 1.0144·$r_{1Z}$ (see Appendix A). Thus, approximating $r_{1}\approx r_{1X}=r_{1Z}$ would result in a negligible error in the calculation of the thermal conductivity $\kappa_{C}$ of the graphene–epoxy nanocomposites. One may expect that the error might be even less in the hybrid graphene@anatase TiO_2_ composite due to the thermal screening of graphene nanoplatelets by titanium dioxide fillers. Therefore, exploiting the case $r_{1X}\neq r_{1Y}$ may seem, at first glance, to not bring any new qualitative conclusions compared to those given above.

We note that the properties of composites appear to depend not only on the size but also on the aspect ratio of the fillers α. It turns out that the interphase layer thicknesses, $h_{1}$ and $h_{2}$, are sensitive to the polymer molecular structure and surface reactivity of the fillers.

One can therefore consider the dependence of $r_{21}$ on the aspect ratio α of the MLGs at varying $h_{1}=h_{2}$, as shown in Figure 4. It is seen that the value of $r_{21}$ depends rather sensitively on the magnitude of α, especially at a high α> 200, when the nanoplatelets can be considered as nanoribbons. In contrast, $r_{21}$ depends only slightly on $h_{1}$ and $h_{2}$ in the range of 10.0 nm $\leq \;h_{1},h_{2}\;\leq$ 30.0 nm.

The results in Figure 4 may help to increase the accuracy in the experimental determination of $r_{21}$. To carry this out properly, one first needs to obtain the distribution function F(α) of the MLGs. Averaging the basic Equation (7) over an interval of F(α) will then give the value of $r_{21}$, which, however, is beyond the scope of this paper.

It should be borne in mind that, to make numerical estimates of the impacts of the interphase layers on the structural characteristics of a three-phase polymer nanocomposite using the relations given in Appendix A, it is necessary, in addition to the empirical parameters $p_{1}$ and $p_{2}$, to predict two independent quantitative parameters of the interphase layers, namely the volume fractions $\varphi_{i1,2}$ and mass densities $\rho_{i1,2}$. Conducting an experimental evaluation of $p_{1,2}$ is a difficult task. They depend on the technique used to prepare the graphene–anatase solution and the adhesion properties of the interacting surfaces. The actual process of self-assembling is stochastic, and different nanoplatelets have different covering degrees and spotted structures. It is evident that the probability of formation of each subsequent anatase layer decreases. Therefore, we may expect an increasing $p_{1,2}$ value when the occupation of platelets increases with the enhancing $C_{2}$ and input either a linear or exponential approximation for the $p_{1,2}\left(C_{2}\right)$ dependence.

We therefore suppose that the assumption of the constant values of $p_{1,2}$ at low loadings will give us essential physical insight. There is, of course, an uncertainty in choosing $p_{1}$ and $p_{2}$. However, we may avoid the ambiguity in the numerical estimates by inputting $p_{1}=p_{2}$ when the geometrical characteristics of the hybrid nanoparticle become independent of both $p_{1}$ and $p_{2}$.

For simplicity, we postulate that the layer’s shape is similar to that of the nanoparticle itself, and the layer’s spatial structure is homogeneous along the surface of the nanoparticle and varies only along the surface’s normal direction. With such an assumption, $\varphi_{i1,2}$ can be determined in terms of the particle dimensions and layer thicknesses $h_{i1,2}$ (see Appendix A). The molecular dynamic simulations testify that the $h_{i}$ values are limited to about $2R_{G}$, with $R_{G}$ being the radius of gyration of the polymer matrix [95,96]. This restriction on $h_{i}$ is a logical consequence of the assumption that the formation of the interphase layer occurs only as a result of the effects of the molecular adhesive forces on the polymer–nanoparticle interface. For instance, for the DGEBA epoxy resins cured with triethylenetetramine (TETA) hardener (C_6_H_18_N_4_), the $R_{G}$ value estimate is 10–14 nm [97]. The radius $R_{G}$ depends on cyclic deformation characteristics such as the operation frequency and loading direction used in the molecular dynamics simulations. Therefore, without the loss of generality, we assume that $h_{1}=h_{2}$ and input $h_{1}=h_{2}=$ 24 nm, supposing an average value of 12 nm for $R_{G}$. The numerical calculations show that an increasing $h_{i}$ value will result in an increase in the nanocomposite’s thermal conductivity $\kappa_{C}$.

As for the values of $\rho_{i1,2}$, they obviously cannot diverse considerably from the epoxy density $\rho_{0}$. The earlier molecular dynamic simulations showed that a range of polymer density fluctuations depends on the nature of the molecular interactions between a filling particle and the polymer macromolecules and it is mainly localized in the close vicinity of the particle of about a few angstroms [98]. On the other hand, using simple considerations based on the free volume concept of the polymer structure allowed us to evaluate the upper limit for the interface layer density for the DGEBA epoxies as 1.064$\rho_{0}$ [4]. Therefore, we assume 0.95$\rho_{0}$ ≤ $\rho_{i1,2}$ ≤ 1.05$\rho_{0}$ to evaluate $\varphi_{i1,2}$. This spread in the value of ρ appears to cause a negligible impact on the calculation results.

For example, if it is assumed that $p_{1}=$ 0.05 and $p_{2}=$ 0.05, one obtains only a 0.0145% variation in the density $\rho_{C}$ of the composite for the case of $\left(C_{1},C_{2}\right)=$ (0.01, 0.01) and 0.035% for $\left(C_{1},C_{2}\right)=$ (0.01, 0.05). For the case of $h_{1}=h_{2}=$ 10 nm and $p_{1}=p_{2}$, the calculated values of the interphase layer factors are $F_{1}=$ 0.4112, $F_{2}=$ 1.744, $F_{3}\left(0.01,0.01\right)=$ 0.1163, and $F_{3}\left(0.01,0.05\right)=$ 0.2654.

Table 4 gives the values of the filler’s ($\varphi_{1,2,3}$) and layer’s ($\varphi_{i1,2}$) volume concentrations, which were calculated by using Appendix A. The difference $∆\varphi_{n}\left(p_{1},p_{2}\right)=\varphi_{n}\left(p_{1},p_{2},\rho_{i1}=1.05\rho_{0},\rho_{i2}=1.05\rho_{0}\right)-\varphi_{n}\left(p_{1},p_{2},\rho_{i1}=0.95\rho_{0},\rho_{i2}=0.95\rho_{0}\right)$ increases with the increasing $p_{1}=p_{2}$ and reaches its maximal value of 1.67·10^−5^ for the case of $p_{1}=p_{2}=$ 1.0, n = 1, $C_{1}=$ 0.01, $C_{2}=$ 0.05. One can see that the interphase areas occupy even greater volume in the composite compared to that in the fillers,
$$\varphi_{i1}+\varphi_{i2}>\varphi_{1}+\varphi_{2}+\varphi_{3},$$
only if $p_{1}=p_{2}=$ 1.0 when $\varphi_{3}=0$, although the above interphase layer thicknesses of $h_{1}=h_{2}=$ 10 nm are not so large.

It can be concluded from these calculations that the chosen sizes of the filler’s loading allow one to avoid the interphase layer overlap and percolation effects. This will be discussed further below.

The overlapping effect can be avoided at low values of $C_{1,2}$. Indeed, the average distance $L_{P}$ between adjacent nanoparticles (with linear dimension $R_{P}$ whose volume fraction $\varphi_{p}$ is distributed homogeneously within the polymer matrix) can be approximated by [99]
(9)$$L_{p}(C_{1},C_{2})=\;R_{p}\left\{{\left[\frac{4\pi }{3\varphi_{p}(C_{1},C_{2})}\right]}^{1/3}-2\right\},$$
where the values of $\varphi_{p}(C_{1},C_{2})$ can be evaluated by using the relations given in Appendix A. By taking $R_{P}=$ 25 nm for the anatase nanoparticles and inputting $\varphi_{p}=\varphi_{1}\left(p_{1}=p_{2}=1.0,C_{1}=0.01,{\;C}_{2}=0.05\right)=$ 5.513 × 10^−3^, we find a minimal value of $L_{2}\approx$ 55 nm. An extreme case of $L_{2,min}=2h_{2}=$ 20.0 nm occurs at $C_{2,max}=$ 0.2407. For graphene nanoplatelets with $R_{P}=\frac{L_{2}}{2}=$ 25 nm, we obtain a similar result,$\;L_{1,min}=2h_{1}=$ 20.0 nm at $C_{1,max}=$ 0.2556.

The percolation thresholds for graphene nanoplatelets occur at the values of $C_{1,PT0}(C_{2})$, which are much lower than those of $C_{1,max}$. Supposing that epoxy is an ideal insulator, the percolation threshold $\varphi_{PT}$ of a composite filled with randomly oriented oblate-shaped ellipsoidal particles is [100]
(10)$$\varphi_{PT}(\alpha )=\;\frac{9S_{33}\left(\alpha \right)[1-S_{33}\left(\alpha \right)]}{-9S_{33}^{2}\left(\alpha \right)+15S_{33}\left(\alpha \right)+2},$$
where $S_{33}$ can be found from Appendix A. For our uncovered graphene nanoplatelets, α= 0.01 and $\varphi_{PT}=\varphi_{PT0}=$ 0.01708, which corresponds to the mass concentrations of $C_{1,PT0}=$ 0.031545 if $C_{2}=$ 0.01 or $C_{1,PT0}=$ 0.030667 if $C_{2}=$ 0.05, provided that $p_{1}=p_{2}=$ 1.0 (i.e., when no assembling occurs) and $\rho_{i1}=\rho_{i2}=\rho_{m}$. To calculate $C_{1,PT0}$, one should use Appendix A and insert $\varphi_{PT0}$ instead of $\varphi_{1}$ and $\rho_{C}(C_{1},C_{2})$ from Appendix A.

Combining the single anatase covering layer with the “hard-spheres” approach yields $\alpha \left(R_{2}\right)=\left(L_{Z}+2R_{2}\right)$/$\left(L_{X}+2R_{2}\right)=$ 150/5100 = 0.02941 and $\varphi_{3PT}\left(\alpha \left(2R_{2}\right)\right)=\varphi_{3PT,max}=$ 0.04718 as extreme values of the percolation threshold for the hybrid MLG@TiO_2_ nanosheets.

In contrast, when the “soft-spheres” model is employed, the anatase covering thickness $h=h\left(C_{1},C_{2}\right)$ is a continuous function of both $C_{1}$ and $C_{2}$, whereas $\alpha =\alpha \left(h\right)=\left(L_{Z}+2h\right)$/$\left(L_{X}+2h\right)$ is a continuous function of h. Then, $\varphi_{3PT}$ can be derived from
$$\varphi_{1}\left(p_{1},p_{2},C_{1},C_{2}\right)=\varphi_{3PT}\left(\alpha (h)\right),$$
where h is a solution of
(11)$$N_{21}\left(p_{1},p_{2},C_{1},C_{2}\right)\cdot v_{2\;}\;=\;\frac{1-p_{2}}{1-p_{1}}\cdot \frac{C_{2}}{C_{1}}\cdot \frac{\rho_{1}}{\rho_{2}}\cdot v_{1}\;=\;\frac{\pi }{6}{\left(L_{X}+2h\right)}^{2}\left(L_{z}+2h\right)-v_{1},$$
with ${v_{1}=v}_{1,1}$ and $v_{1}=v_{1,2}$ given by Appendix A for parallelepiped- and spheroid-shaped nanoplatelets, respectively. The numerical solution of Equation (11) for $p_{1}=p_{2}=$0 (complete assembling) gives the set of values of the percolation thresholds presented in Table 5 for the concentrations that are reachable in our experimental samples.

Interestingly, varying the mass density $\rho_{i1,2}$ by 5% around the value of $\rho_{m}$ at $C_{1}=$ 0.01 and $C_{2}=$ 0.05 caused the percolation threshold to change to not more than 0.01% and 0.023% for the parallelepiped- and spheroid-shaped configurations, respectively. It is apparent, therefore, that the effects of the interphase layer overlaps, and percolation cannot be deduced from the experimental κ data.

### 2.5. Discussing the Loading Dependencies

Figure 5 shows the loading dependencies of the thermal conductivities of the MLG@TiO_2_-epoxy nanocomposites calculated for different cases of interphase interactions. Figure 5a corresponds to the case when the filler–epoxy macromolecule interactions are neglected so that interphase layers cannot be formed. If the fillers remain unassembled, curve 1 is reproduced. In the opposite case of the fillers that are fully assembled in hybrid particles, curves 2 and 3 are obtained. A summary of the data in Figure 5a shows the following trends:

(1) $\kappa_{C}$ increases with the increasing $C_{2}$ regardless of whether or not the assembling effect occurs;

(2) The particle assembling decreases $\kappa_{C}$, and the greater the assembling degree $p_{1}+p_{2}$, the smaller the observed $\kappa_{C}$ value;

(3) $\kappa_{C}$ decreases with the increasing graphene–anatase thermal boundary resistance $r_{21}$, i.e., with the weakening of the MLG-TiO_2_ interphase interaction.

It is also interesting to note that $\kappa_{C}$ increases with the increasing $\kappa_{g11}$, $\kappa_{g33}$ or $\kappa_{2}$ values, as expected.

Figure 5b represents the case when the fillers interact with the epoxy macromolecular chains, and thus, interphase layers are formed. Curve 4, similarly to curve 1, corresponds to unassembled fillers, while curves 5 and 6 correspond to fully assembled fillers. One can see that including an interphase layer results in a decreasing $\kappa_{C}$ value. The other trends deduced from Figure 5a are also seen in Figure 5b.

Figure 6 shows the dependencies of $\kappa_{C}$ in the composite with fully assembled graphene and anatase particles on the thermal boundary resistance at the graphene–anatase interface ($r_{21}$) computed at the fixed values of the thermal boundary resistance at the anatase–epoxy interface ($r_{2}$). As expected, a decreasing $r_{21}$ value results in a rapid increase in $\kappa_{C}$. Instead, it is useful to note that decreasing the $r_{2}$ value (or increasing either $C_{2}$ or $\kappa_{g11}$) increases the ratio $\kappa_{C}\left(r_{21min}\right)/\kappa_{C}\left(r_{21max}\right)$, e.g., 1.43 for $r_{2}=$ 1.0 × 10^−7^ m^2^ K W^−1^ (curves 3 and 9 in Figure 6) against 6.51 for $r_{2}=$ 1.0 × 10^−9^ m^2^ K W^−1^ (curve 10).

Figure 7 shows similar dependencies upon $r_{2}$ at fixed values of $r_{21}$. The computed results show a similar trend, as observed in Figure 6. Thus, an increased $\kappa_{C}$ value is observed upon the decreasing $r_{2}$ value, while an increased $\kappa_{C}\left(r_{2min}\right)/\kappa_{C}\left(r_{2max}\right)$ value is realized with the decreased $r_{21}$, increased $C_{2}$, or increased $\kappa_{g11}$, from 1.55 for $r_{21}=$ 1.0 × 10^−7^ m^2^ K W^−1^ (curves 3 and 9 in Figure 7) to 7.76 for $r_{21}=$ 1.0 × 10^−9^ m^2^ K W^−1^ (curve 10). Remarkably, a sensitivity of $\kappa_{C}$ to varying $\kappa_{g11}$ increases with the increasing $C_{2}$.

A general expression for the effective thermal conductivity $\kappa_{f,eff}$ of a particle embedded into a polymer matrix has been obtained in [101] in the effective media approximation. We have used this expression to write $\kappa_{f,eff}$ for free anatase particles ($\kappa_{2,eff}$, Appendix A), free graphene platelets ($\kappa_{g11,eff}$ and $\kappa_{g33,eff}$, Appendix A), and hybrid graphene@anatase particles ($\kappa_{h,eff}$, Appendix A).

Figure 8 shows the dependencies of normalized effective thermal conductivities of filling particles embedded into epoxy on correspondent thermal boundary resistances.

Thus, as expected, the effective thermal conductivities of the filling particles increase with the decreasing r value because this decrease is accompanied by an increased conductivity $\kappa_{int}$ of the interphase layers given by Equation (6).

It should be mentioned that the curves presented in Figure 6 and Appendix A predict the bounds for $\kappa_{C}$, where they can be tailored by varying the thermal boundary resistances, $r_{2}$ and/or $r_{21}$ (in the case of complete self-assembling), if both $C_{1}$ and $C_{2}$ remain unchanged.

Evidently, the value of the thermal boundary resistance at the filler–polymer interface ($r_{1}$ and $r_{2}$ in our case) can vary via the chemical modification of the surfaces of the filling particles. This surface functionalization involves the conjugation of suitable functional molecular groups with the active surface sites of the particles. Generally speaking, the functionalization of fillers with the requisite moieties depends on the chemistry of the fillers. Recently, efficient techniques for the functionalization of carbon nanomaterials were developed (see refs. [102,103] and references therein), the most important of which may be classified as either covalent or non-covalent functionalization.

Covalent functionalization alters the state of bond connectivity between the fillers and a host polymer matrix. The functional groups form the covalent linkage between the polymer’s macromolecules and the dangling carbon bonds located on the skeletons of graphene nanoplatelets (mainly on its edges and in structural defects). For example, the covalent interactions of oxidized graphene’s basal and lateral surfaces with epoxy are dominated by hydroxyl, epoxy, and carboxylic molecular groups [104]. As a result of functionalization, the translational symmetry of the outer graphene nanosheets of MLG-nanoplatelets is disrupted by changing the ${sp}^{2}$ carbon atoms to ${sp}^{3}$ carbon atoms, and both the electronic and phonon transport properties of the sheets are influenced.

Moreover, an enhanced conjugation between fillers and a host polymer matrix can happen through noncovalent interactions by attaching some specific secondary ligands on the filler’s surface. Various approaches of filler-to-ligand noncovalent binding include electrostatic interaction, π-π stacking, and H-bonding [103]. For MLG fillers, noncovalent functionalization, in contrast to covalent functionalization, has no effect on the π-π stacking of the multilayered nanoplatelet [105].

The grafted individual moieties or ligand-moiety chains in a functionalized filler can act as heat transfer channels between the filler and a host polymer matrix, decreasing phonon scattering at this interface and thus reducing the thermal boundary resistance r or increasing the heat exchange between the filler and the matrix [106,107]. In other words, grafted species increase the overlapping phonon density of states of the functionalized filler and the polymer matrix.

On the other hand, the surface functionalization breaks the regular surface structure of the fillers and reduces the intrinsic mean free path length of the phonons, acting as additional scattering centers for the phonons. Molecular dynamic simulations showed that the in-plane thermal conductivity of infinitely long graphene nanoribbons ($\kappa_{\infty }$) decreases sharply with the increasing the grafting density $g_{d}$ [107]. Specifically, $\kappa_{\infty }$ reduces to about 14% from that for ungrafted graphene at $g_{d}=$ 1.6%. Furthermore, the thermal conductivity of grafted graphene at $g_{d}=$ 12.5% is only 3.86% of that for ungrafted graphene. Qualitatively similar effects may be expected for graphene functionalized by anatase nanoparticles.

Therefore, the impact of the filler functionalization on the composite’s thermal conductivity appears to be twofold due to the competition between these two phenomena, an enhanced thermal conductance at the interface and the reduced intrinsic thermal conductivity of the filler. A certain optimal value $g_{d,opt}$ may be expected for filling particles, which is required to maximize the ratio $\kappa_{C}\left(g_{d,opt}\right)/\kappa_{C}\left(g_{d}=0\right)$.

We can further extend the relevance of this model by providing the way in which these functionalization mechanisms contribute together to produce effective thermal conductivity. One possible way to account for the functionalized graphene is to consider the intrinsic thermal conductivities of graphene ($\kappa_{g11}$, $\kappa_{g22}$, $\kappa_{g33}$) and anatase ($\kappa_{2}$) and the corresponding thermal boundary resistances ($r_{1X},r_{1Y},r_{1Z},r_{2}$, $r_{21}$) as variables that are dependent on the grafting densities $g_{d1}$ and $g_{d2}$. Thus, the resulting set of functions including $\kappa_{g11}\left(g_{d1}\right),$ $\kappa_{g22}\left(g_{d1}\right),$ $\kappa_{g33}\left(g_{d1}\right),$ $\kappa_{2}\left(g_{d2}\right),$ $r_{1X}\left(g_{d1}\right),$ $r_{1Y}\left(g_{d1}\right),$ $r_{1Z}\left(g_{d1}\right),$ $r_{2}\left(g_{d2}\right),$ and $r_{21}\left(g_{d1}{,g}_{d2}\right)$ should be prescribed. Very little information related to these functions is available. For example, $\kappa_{g11}$ and $r_{1}$ were empirically related to $g_{d1}$ for graphene nanoplatelets embedded in the polyamide-6,6 matrix [107], although the dependences look somewhat intricate.

It should also be noted that the synergetic effects of functionalization or assembling fillers can be more pronounced when the filler surfaces are subjected to preliminary purification. It is well known that different atomic and molecular impurities conjugated with filler surfaces have detrimental influences on the intrinsic and interfacial thermal properties of the filler [108,109,110,111,112]. In particular, the trace metallic and metalloid impurities [108], acidic residues [109], hydrocarbones [110], and even silicon [111] are ubiquitous on graphene surfaces.

In this study, no effort was undertaken to intentionally purify or functionalize the fillers, neither graphene nor anatase, because the main aim of this study is to reveal the thermal synergetic effects of assembling graphene and anatase. Therefore, we have no need to cater for laborious and time-consuming techniques for the purification surface of the fillers and the subsequent quantitative estimation of the type and amount of residual functional groups.

Finally, we compare the proposed model predictions and experimental results shown in Figure 1a. The open and closed triangles give the distributions of the calculated values of the thermal conductivity in our composites at different key parameters of the model. Varying the aspect ratio α from 0.005 to 0.02, the varying thermal boundary resistances $r_{2}$ and $r_{21}$ from 10^−5^ to 10^−10^ m^2^∙W^−1^∙K^−1^, and varying the thicknesses of the interphase layers $h_{1}=h_{2}$ from 10 to 30 nm yield a fair agreement of the calculated and experimental results. The bottom mesh of points (open triangles) characterizes the hypothetical situation of extremely small contact thermal resistances between the matrix and the fillers (minimum phonon exchange between the phases of the composite), whereas the upper mesh (closed triangles) corresponds to a strong interphase interaction with a significant overlap of the phonon density spectra of the fillers and the matrix. These calculations were performed in the soft spheres approximation (complete coverage of the surfaces of MLGs with TiO_2_) for the complete self-assembling (at $p_{1}=p_{2}=$ 0) of graphene and anatase in an epoxy polymer matrix. In this case, the variation in the TiO_2_ diameter does not affect the calculation results. It is also important to note that the effect of incomplete coverage of the graphene surface with anatase (observed in Figure 4) on the calculated thermal conductivity is very small at the small numbers of filling particles $C_{1}$ and $C_{2}$ that were used in the experiments (last three paragraphs in Appendix A).

## 3. Materials and Methods

MLG nanoplatelets for our experiments were prepared from the flakes of thermally expanded graphite by using the electrochemical technique described by Xia et al. [113]. The structural properties of the nanoplatelets were tested by using X-ray diffraction (XRD) spectroscopy as described in our previous report [14]. A distinct difference between the XRD patterns of the initial graphite flakes and graphene sheets was observed [14]. Moreover, the Raman spectra of single-layer graphene particles and similarly fabricated samples of multilayer graphene particles were compared by Gorelov et al. [10], illustrating that our multilayer graphene nanoplatelets are composed of loosely bound graphene sheets. One can therefore suggest that the fabricated particles are multilayered graphene nanoplatelets rather than exfoliated graphite platelets.

To prevent the resulting MLG material from oxidizing, it was kept as the suspension. The particles were about 5 × 5 μm^2^ in in-plane dimensions and 790 m^2^/g in specific surface area. The value of the specific surface area was determined by measuring the amount of physically adsorbed nitrogen gas from adsorption–desorption isotherms according to the standard Brunauer, Emmett, and Teller (BET) method [114] by using an Autosorb Station 3 apparatus of Quantachrome Instruments (Boynton Beach, FL, USA).

The TiO_2_-anatase nanoparticles were deposited on the MLG by adding them into the initial ethanol-based suspension of the MLG before its ultrasonic treatment. The specific surface area of the anatase particles was ~1500 m^2^/g.

The commercially available CHS-EPOXY 520 (SpolChemie a.s., Usti nad Labem, Czech Republic) DGEBA-epoxy resin based on Bisphenol A diglycidyl ether (DGEBA), of epoxy group content (E-Index) 5.21–5.50 mol/kg, with an EEW (Epoxy Equivalent Weight) of 182–192 g/mol was used as the neat resin. Diethylenetriamine (abbreviated as Dien or DETA) was used as a curing agent. DETA is a nitrogen-containing organic compound with the formula HN(CH_2_CH_2_NH_2_)_2_ [115]. The epoxide to the hardener mass ratio was kept constant at 7:1. Details of the curing process can be found elsewhere [116].

The MLG mass loading values ($C_{1}$) for the nanocomposites were 0.5%, 1%, 2%, and 5%, whereas the TiO_2_ mass loading values ($C_{2}$) were taken to be 0.5%, 1%, and 5%. To prepare the nanocomposites of the prescribed content $\left(C_{1},C_{2}\right)$, the following procedure was used: The epoxide olygomer to the hardener mass ratio ($R_{OH}$) was kept constant at 7:1, i.e., $R_{OH}=\frac{m_{0}}{m_{h}}=$ 7. We hence started by taking the epoxide olygomer portion with the mass $m_{0}=$ 105 g poured out into a quartz glass. Then, we added the fillers in the following two steps: first, we added the 10% graphene–ethanol suspension portion with the graphene mass
(12)$$m_{g}=\frac{C_{1}m_{e}}{1-C_{1}-C_{2}}$$
and second, we added the dry anatase powder with the mass $m_{a}$ given by
(13)$$m_{a}=\frac{C_{2}m_{e}}{1-C_{1}-C_{2}},$$
where $m_{e}=m_{0}+m_{h}=m_{0}+\frac{1}{7}m_{0}=$ 120 g.

The as-prepared liquid composites were ultrasonically mixed until homogeneous suspensions were obtained. Then, they were stored in a vacuum chamber to remove ethanol and air bubbles. Finally, a hardener portion of $m_{h}=$ 15 g was added to the treated compound, and the resultant mixture was mixed. After adding the hardener to the suspensions, their further polymerization occurred at room temperature for 72 h. Once again, the suspensions were stored in a vacuum for the first two hours at a reduced pressure of about 10^−2^ mbar to remove all the emerging gaseous constituents and thus prevent the formation of pores within the samples.

The morphology features of the nanocomposite were investigated with scanning electron microscopy (SEM) using a JEOL JSM-6490 microscope and with transmission electron microscopy (TEM) using a JEOL JEM-2100 microscope. Figure 9 shows the SEM image of the TiO_2_ particles decorating the graphene nanosheets (a), and TEM images of a hybrid graphene@anatase TiO_2_ nanocomposite are shown in (b) and (c). It is seen in Figure 9a that the TiO_2_ particles are well dispersed over the MLG surfaces. The high-resolution TEM image shown in Figure 9c displays a lattice fringe of about 0.33 nm, which can be attributed to the anatase (101) plane [117].

It was revealed by the microscopy that the TiO_2_ particles have a diameter of ~30–70 nm, while the thickness of the MLG nanoplatelets ranges from about 30 to 150 nm. One needs to keep in mind that these dimensions are randomly distributed through irregular MLG@TiO_2_ platelets. So, one needs to choose the values of the MLG and TiO_2_ thicknesses (both are taken to be 50 nm above), and then obtain the numerical continuation solution branches in κ upon varying parameters. Some results describing varying $\kappa_{C}$ upon varying MLG thickness are given in Appendix A.

Here, we used the two-probe measurement method to measure thermal conductivity [118]. Our automatic measuring system is schematically sketched in Figure 10. The sample was placed between two copper discs (heater and thermostat in Figure 10). An electric resistance served as a heater, and the power P released on it was determined with the electric current and voltage applied to the resistance. Additionally, the heat flux through the sample is Q=P/S, where S is the top and bottom surface area of the sample. A thermostat was used to keep the temperature of the top surface of the sample constant.

By maintaining the stationary heat flux, one can write the following relation to evaluate the thermal conductivity [119]
(14)$$\kappa =\frac{QL}{T_{A}-T_{B}},$$
where L is the distance between the sensors, and $T_{A}$ and $T_{B}$ are the temperatures measured by sensors A and B, respectively.

The samples were cylindrically shaped with a diameter of 12.5 mm and a height of 6 mm. Two holes with diameters of 1 mm were drilled in the samples from the top and bottom surfaces to the center, in which the temperature sensors were placed (A and B in Figure 10). After that, the holes were filled with epoxy resin for better thermal contact between the sample and the temperature sensors. BAP64-02 NXP diodes were used as temperature sensors. The diodes had rather small dimensions of 1.2 × 0.8 × 0.6 mm^3^, which made it possible to place them inside the sample without difficulty.

We believe the diode temperature sensor was small enough to give rise to correct measurements since the ratio of the cross section of the sensor to that of the sample is less than 1%. There is little work reported on the usage of diode sensors to obtain thermal conductivity measurements of epoxy systems [120]. It may therefore be expected that the diode cannot lead itself to significantly interrupt the temperature field of the measurement.

Furthermore, the main channel of heat delivery to the pn junction is not its plastic cover, but rather the metal terminals to which the electrical lead wires are soldered. The thermal resistance to energy incoming from the junction to the soldering point is 85 K/W according to the diode performance specifications. Hence, though the diode is too big to be ignored and inevitably affects the characteristics of the temperature field, it is immediately obvious that the temperature sensor here is formed by the metal leads, which are completely immersed in the epoxy matrix. The surface area of the lead wires is comparable to the contact area of a thermocouple that is more commonly used to minimize the sensor influence. It is also obvious that the thermal resistance between the metal leads and epoxy resin will be more or less the same in both cases.

Moreover, temperature measurements are significantly affected by the thermal time constant. The thermal time constant of the system is the time required to come to within 1/e (≈37%) of the fixed value after a step thermal disturbance of the system [119]. The thermal time constant of the temperature sensor must be significantly less than the thermal time constant of the sample. Thus, the thermal resistance between the sensor and the sample will not significantly affect the measurement of the sample temperature. The thermal time constant of the diode is
(15)$$\tau_{t}=R_{t}C_{t},$$
where $R_{t}$ is the resistance of the regions next to the junction, and $C_{t}=m_{d}C_{d}$ is the lumped thermal capacitance of the diode with the mass $m_{d}$ and specific heat $C_{d}$. Taking $R_{t}=$ 85 K/W given above and $C_{d}=$ 800–1200 J·kg^−1^·K^−1^ for the diode molding compound yields $\tau_{t}$ on the order of hundreds of ms in the worst case. Essentially, our measurements were carried out in a quasi-static regime such that, after setting the required temperature on the controller, the measuring system waited for dozens of minutes to achieve temperature stabilization. This waiting time depended on the temperature and was set to provide the rate of heating of the sample of less than 2 K/hour. This waiting time exceeded the thermal time constant of the diode. The sensors therefore did not appear to have an appreciable effect on the temperature measurements.

We also implemented a 2D finite element method (FEM) to model the temperature distribution using the COMSOL 5.3a and typical thermal conductivity parameters for diode materials, which are κ= 400 W/mK for copper and κ= 0.58–0.67 W/mK for the diode forming mass [121]. The simulation results are given in Figure 11 and Figure 12. It is seen in Figure 11 that the temperature field around the diodes is not significantly distorted.

The temperature variation across the dashed line at x= 0.7 mm in Figure 11c from the bottom to the top sample surface is shown in Figure 12. The dashed line simulates the variation in the absence of the diode sensors, whereas the solid line approximates the one with the embedded diodes. The dashed line in Figure 11c passes through the centers of the diode electrodes, which are heat-sensitive elements in our case. One can therefore see a perturbation in the temperature field at around y≈±1 mm just near the metal electrodes. This is due to the much higher thermal conductivity of the electrode than the epoxy resin. However, at the outer side of the electrodes (at y≈±1.15 mm in Figure 12), the solid- and dashed-line temperatures are practically identical. This ensures that the experimental temperature data obtained from these measurements are accurate and reliable when taking into account the thicknesses of the electrodes and when measuring the inter-sensor distance between their external edges.

Other factors to be taken into account when assessing the accuracy of the measurements are as follows: Copper wires with a thickness of 50 μm and a length of 100 mm were used to feed the diodes. Their thermal resistance is much higher than the thermal resistance of the sample, allowing for a heat leakage to be quenched. The epoxy resin without hardener was used to give a better thermal contact between the sample, heater, and cooler. A plastic screw defining a point contact constriction with a heat-insulating fastener was used to clamp the sample, which yielded a heat loss of less than 1%. With the above assumptions, the errors in the measured thermal conductivities do not exceed about 5%.

The sample was placed in a closed-cycle cryostat (CS204, Advanced Research Systems) that was used to vary its temperature. The temperature in the cryostat was controlled using a Lake Shore 331S controller through the computer’s RS232 digital port. Measurements were carried out in the temperature range from 40 to 300 K.

## 4. Conclusions

A theoretical model based on the effective medium approximation was developed to calculate the loading dependence of the thermal conductivity κ in hybrid graphene–polymer nanocomposites and evaluate its sensitivity to the geometrical and material parameters of filling particles and the host polymer matrix, filler’s mass concentration, self-assembling degree, and Kapitza thermal boundary resistances at emerging interfaces. The hybrid graphene@anatase nanosheets were modeled as sandwich-like structures of either parallelepiped- or spheroid-like configurations, where the geometrical and material characteristics of the anatase layer were treated in a continuous media approximation for two extreme cases representing anatase nanoparticles as either soft or hard solid spheres. The differences between the model results obtained with the above two configurations increased with the decreasing the nanosheet’s aspect ratio. The model was also capable of predicting the $\kappa \left(T\right)$ dependencies in the nanocomposites if the thermal parameters of the constituent phases and the thermal boundary resistances were known. The soft-sphere approach and related concept of the effective thickness allowed us to simplify the simulations and describe the interface thermal transport in the hybrid graphene@anatase nanosheets via the phenomenological thermal boundary resistance.

We also measured the temperature dependence of the thermal conductivity of epoxy nanocomposites filled with self-assembled hybrid nanoparticles composed of multilayered graphene nanoplatelets and anatase-TiO_2_ nanoparticles within a temperature range from 40 K to 320 K. When the concentration of anatase in the composites increased, a saturation tendency in the $\kappa \left(T\right)$ dependence was observed above ≈150 K, which was possibly due to a saturation of the thermal conductivity of the anatase. A fair agreement between the calculated and measured room-temperature thermal conductivity was obtained.

The numerical results clearly illustrate the thermal screening effect of a low-conductive anatase layer that covers high-conductive graphene nanoplatelets. The observed growing thermal conductivity after raising the temperature is not only related to the increasing intrinsic thermal conductivities of both the fillers and host epoxy but is also subject to temperature-induced variations in the thermal boundary resistances.

An effective tailoring of κ is possible only if the anatase-to-graphene mass ratio does not exceed a certain bound, which corresponds to the sheer covering of graphene nanoplatelets by anatase nanoparticles. It is furthermore expected that κ will increase further when decreasing the thermal boundary resistances at the anatase–graphene interface. The latter can be realized by using more sophisticated techniques to chemically graft anatase to the graphene surface. However, when the concentration of hybrid nanosheets approaches its percolation thresholds with an increasing anatase-to-graphene concentration ratio, the temperature dependence of the composite’s thermal conductivity weakens. It is therefore suggested that varying the filler concentration that covers the graphene nanoplatelets offers interesting opportunities of applied relevance in the area of self-assembling engineering of graphene surfaces.

## Figures and Tables

**Figure 1 molecules-28-07343-f001:**
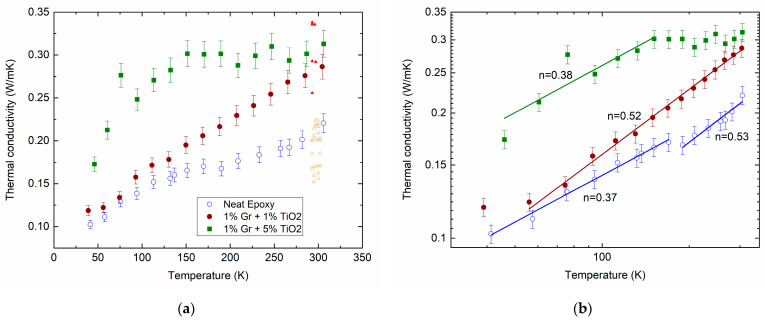
Measured temperature dependencies of κ for the neat epoxy (open circles) and epoxy-MLG@TiO_2_ nanocomposites with different mass loading ratios for MLG ($C_{f1}$) and TiO_2_ ($C_{f2}$): $C_{f1}:C_{f2}=$ 0.01:0.01 (closed circles) and 0.01:0.05 (squares) on lin–lin (**a**) and log–log (**b**) scales. Lines in (**b**) are fitted to a power law $\kappa \left(T\right)\propto T^{n}$ with the exponent values n. Triangles in (**a**) are the calculated results at 300 K (for a detailed description, see Section 2.5). They are arbitrarily shifted along the temperature axis for better comparison.

**Figure 2 molecules-28-07343-f002:**
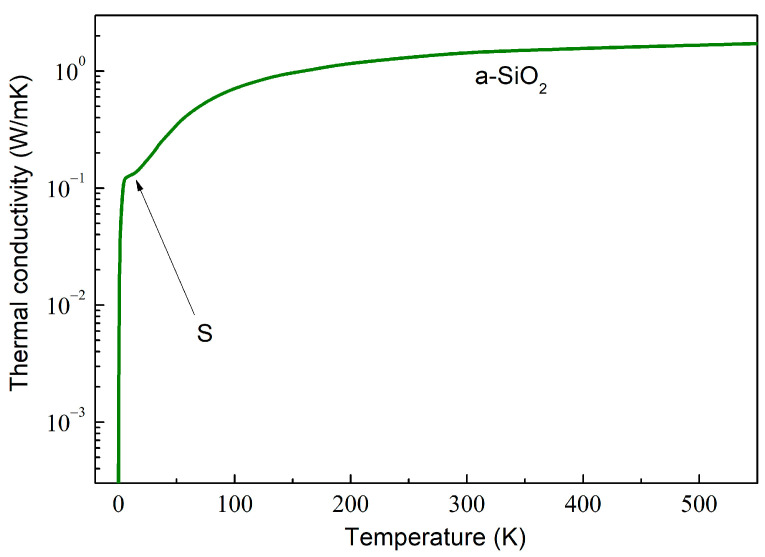
A log–lin plot of the thermal conductivity of amorphous SiO_2_, calculated by using the data obtained by Goodson et al. [65].

**Figure 3 molecules-28-07343-f003:**
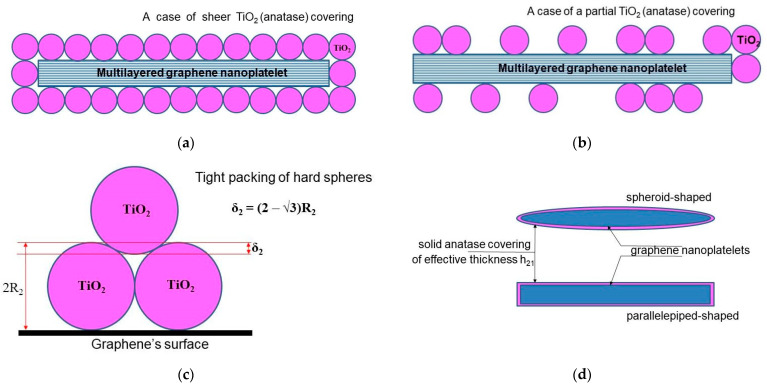
Regular geometrical architectures of hybrid MLG@TiO_2_ nanosheets configured in a sandwich-like structure: (**a**) parallelepiped-shaped graphene nanoplatelet covered entirely with hard spherical anatase nanoparticles; (**b**) parallelepiped-shaped graphene nanoplatelet covered partially with hard spherical anatase nanoparticles; (**c**) geometrical effect of tight packing of hard spheres on a flat surface; (**d**) soft-sphere approach to model anatase wrapping of graphene nanoplatelet as a solid layer with the effective thickness of $h_{21}$.

**Figure 4 molecules-28-07343-f004:**
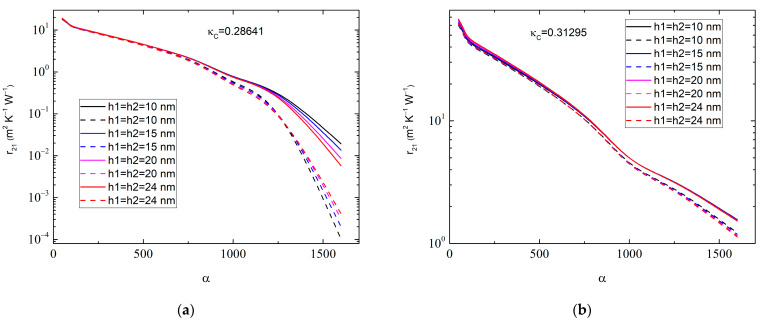
Graphene–anatase thermal boundary resistance $r_{21}$ as a function of graphene aspect ratio α for (**a**) $C_{1}:C_{2}=$ 0.01:0.01, $\kappa_{C}=\kappa_{C,E1}=$ 0.2864 W∙m^−1^∙K^−1^ and (**b**) $C_{1}:C_{2}=$ 0.01:0.05, $\kappa_{C}=\kappa_{C,E2}=$ 0.31295 W∙m^−1^∙K^−1^. Different colors correspond to various values of interphase layer thicknesses: $h_{1}=h_{2}=$ 10 nm (black curves), 15 nm (blue curves), 20 nm (magenta curves), and 24 nm (red curves). Solid lines correspond to $r_{2}=$ 1.0 × 10^−9^ m^2^∙K^−1^∙W^−1^ and dashed lines correspond to $r_{2}=$ 1.0 × 10^−8^ m^2^∙K^−1^∙W^−1^. The MLG thickness is $L_{Z}=$ 50.0 nm, whereas $L_{X}=L_{Y}=\alpha L_{Z}$.

**Figure 5 molecules-28-07343-f005:**
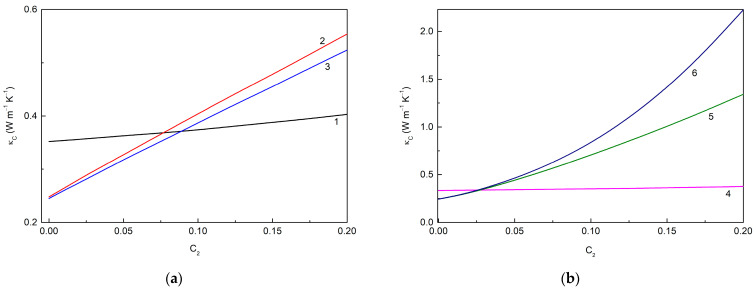
Loading dependencies of the thermal conductivity $\kappa_{C}$ of MLG@TiO_2_-epoxy nanocomposites calculated for different cases of interphase interactions (**a**) without interphase layers (($h_{1}=0,h_{2}=0$), $\kappa_{g11}=$ 600.0 W m^−1^ K^−1^, $\kappa_{g33}=$ 6.0 W m^−1^ K^−1^: curve 1—non-assembled particles ($p_{1}=p_{2}=$ 1.0) ($\kappa_{C}$ is independent of $r_{21}$); curve 2—fully assembled particles ($p_{1}=p_{2}=$ 0), $r_{21}=$ 1.3536 × 10^−7^ m^2^ K W^−1^; curve 3—fully assembled particles ($p_{1}=p_{2}=$ 0), $r_{21}=$ 1.5310 × 10^−7^ m^2^ K W^−1^) and (**b**) with interphase layers (($h_{1}=h_{2}=$ 10 nm), $\kappa_{g11}=$ 600.0 W m^−1^ K^−1^, $\kappa_{g33}=$ 6.0 W m^−1^ K^−1^: curve 4—non-assembled particles ($p_{1}=p_{2}=$ 1.0); curve 5—fully assembled particles ($p_{1}=p_{2}=$ 0), $r_{1}=$ 3.5 × 10^−9^ m^2^ K W^−1^, $r_{2}=$ 1.0 × 10^−8^ m^2^ K W^−1^, $r_{21}=$ 1.5310 × 10^−7^ m^2^ K W^−1^; curve 6—fully assembled particles ($p_{1}=p_{2}=$ 0), $r_{1}=$ 3.5 × 10^−9^ m^2^ K W^−1^, $r_{2}=$ 1.0 × 10^−9^ m^2^ K W^−1^, $r_{21}=$ 1.3536 × 10^−7^ m^2^ K W^−1^).

**Figure 6 molecules-28-07343-f006:**
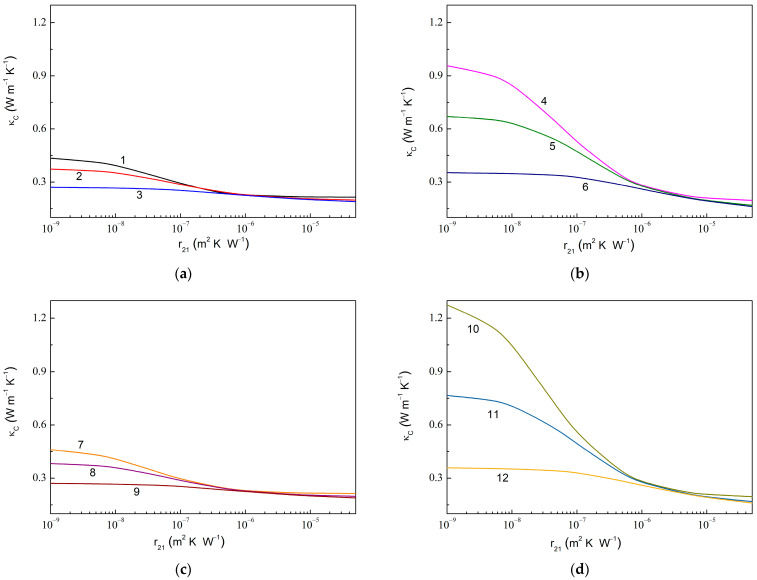
Thermal conductivity of MLG@TiO_2_-epoxy nanocomposites vs. thermal boundary resistance $r_{21}$ at graphene–anatase interface calculated at fixed values of the thermal boundary resistance $r_{2}$ at anatase–epoxy interface for the case of fully assembled graphene and anatase particles at (**a**) $C_{1}=$ 0.010, $C_{2}=$ 0.010, $\kappa_{g11}=$ 600.0 W m^−1^ K^−1^, $\kappa_{g33}=$ 6.0 W m^−1^ K^−1^; (**b**) $C_{1}=$ 0.010, $C_{2}=$ 0.050, $\kappa_{g11}=$ 600.0 W m^−1^ K^−1^, $\kappa_{g33}=$ 6.0 W m^−1^ K^−1^; (**c**) $C_{1}=$ 0.010, $C_{2}=$ 0.010, $\kappa_{g11}=$ 1200.0 W m^−1^ K^−1^, $\kappa_{g33}=$ 6.0 W m^−1^ K^−1^; and (**d**) $C_{1}=$ 0.010, $C_{2}=$ 0.050, $\kappa_{g11}=$ 1200.0 W m^−1^ K^−1^, $\kappa_{g33}=$ 6.0 W m^−1^ K^−1^. The values of $r_{2}$ are 1.0 × 10^−9^ m^2^ K W^−1^ (curves 1, 4, 7, and 10), 1.0 × 10^−8^ m^2^ K W^−1^ (curves 2, 5, 8, and 11), and 1.0 × 10^−7^ m^2^ K W^−1^ (curves 3, 6, 9, and 12).

**Figure 7 molecules-28-07343-f007:**
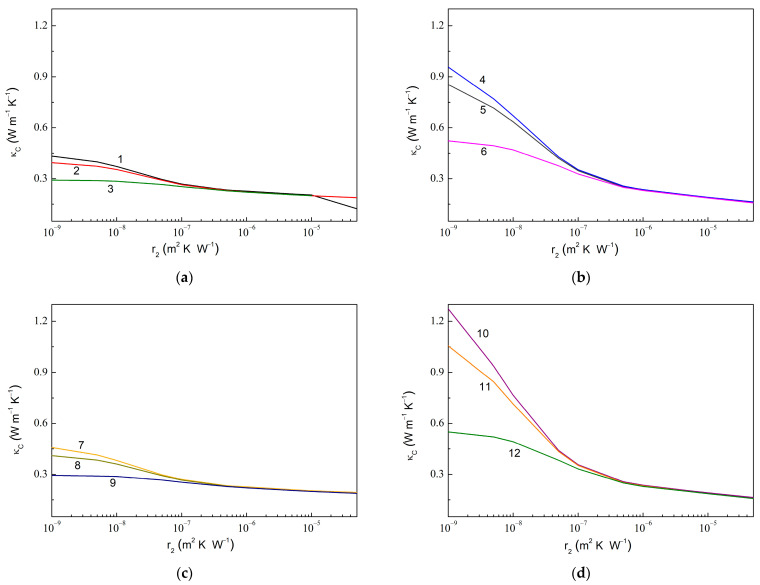
Same dependencies as in Figure 6 but upon varying $r_{2}$ and fixed $r_{21}$. (**a**) $C_{1}=$ 0.010, $C_{2}=$ 0.010, $\kappa_{g11}=$ 600.0 W m^−1^ K^−1^, $\kappa_{g33}=$ 6.0 W m^−1^ K^−1^; (**b**) $C_{1}=$ 0.010, $C_{2}=$ 0.050, $\kappa_{g11}=$ 600.0 W m^−1^ K^−1^, $\kappa_{g33}=$ 6.0 W m^−1^ K^−1^; (**c**) $C_{1}=$ 0.010, $C_{2}=$ 0.010, $\kappa_{g11}=$ 1200.0 W m^−1^ K^−1^, $\kappa_{g33}=$ 6.0 W m^−1^ K^−1^; and (**d**) $C_{1}=$ 0.010, $C_{2}=$ 0.050, $\kappa_{g11}=$ 1200.0 W m^−1^ K^−1^, $\kappa_{g33}=$ 6.0 W m^−1^ K^−1^. The values of $r_{21}$ are 1.0 × 10^−9^ m^2^ K W^−1^ (curves 1, 4, 7, and 10), 1.0 × 10^−8^ m^2^ K W^−1^ (curves 2, 5, 8, and 11), and 1.0 × 10^−7^ m^2^ K W^−1^ (curves 3, 6, 9, and 12).

**Figure 8 molecules-28-07343-f008:**
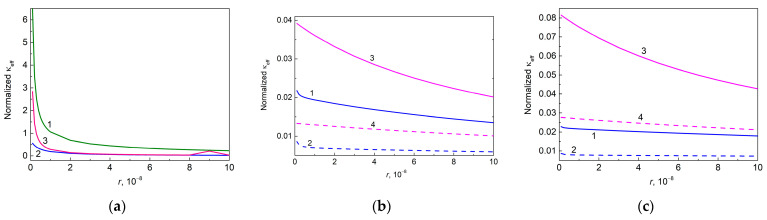
Normalized thermal conductivity of filling particles embedded into epoxy as a function of thermal boundary resistance, calculated by using Appendix A, (**a**) for free particles of anatase (curve 1, $\kappa_{a,eff}/$0.455, $r=r_{2}$) and graphene (curve 2, $\kappa_{g11,eff}/$600.0, $r=r_{1}$) (curve 3, $\kappa_{g33,eff}/$6.0, $r=r_{1}$); (**b**) for hybrid graphene@anatase particles ($p_{1}=p_{2}=0$, $C_{1}=$ 0.010, $C_{2}=$ 0.010): $\kappa_{h11,eff}/$600.0 (curves 1 and 2) and $\kappa_{h33,eff}/$6.0 (curves 3 and 4); and (**c**) for hybrid graphene@anatase particles ($p_{1}=p_{2}=0$, $C_{1}=$ 0.010, $C_{2}=$ 0.050): $\kappa_{h11,eff}/$600.0 (curves 1 and 2) and $\kappa_{h33,eff}/$6.0 (curves 3 and 4). Solid and dashed lines in (**b**,**c**) correspond to $r_{21}=$ 1.0 × 10^−7^ m^2^ K W^−1^ and 3.0 × 10^−7^ m^2^ K W^−1^, respectively.

**Figure 9 molecules-28-07343-f009:**
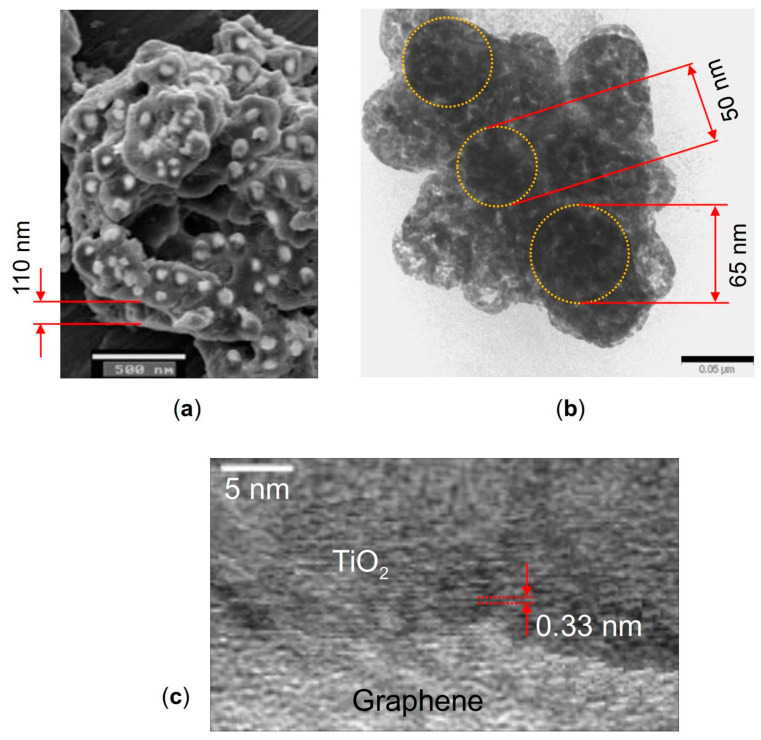
(**a**) SEM image of anatase TiO_2_ particles decorating graphene nanosheets, (**b**) TEM image of the graphene flake with TiO_2_ particles on it marked by dotted circles, and (**c**) high-resolution TEM image of a hybrid graphene flake@anatase TiO_2_ nanocomposite. The scale bars are 500 nm in (**a**) and 50 nm in (**b**).

**Figure 10 molecules-28-07343-f010:**
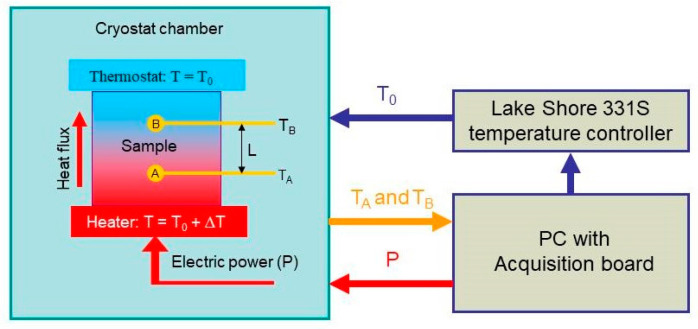
Schematics of the setup used for the thermal conductivity measurements. A and B are the temperature sensors embedded into the sample. L is the distance between the sensors. Thermostat is formed by mounting the top sample edge onto the cold head of the cryostat used (T-controlled heat sink).

**Figure 11 molecules-28-07343-f011:**
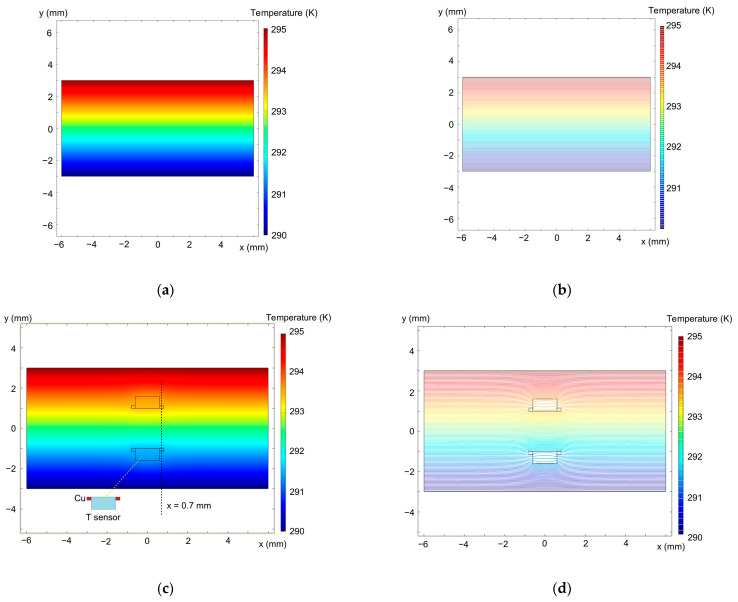
Simulated temperature distribution in the sample plane across the diode sensors in full color (left-hand images) and contour (right-hand images) plots. (**a**,**b**)—without the diodes; (**c**,**d**)—with the temperature sensors embedded into the sample (see Figure 10). The upper sample surface is heated. The dashed line at x= 0.7 mm in (**c**) shows the temperature distribution given in Figure 12.

**Figure 12 molecules-28-07343-f012:**
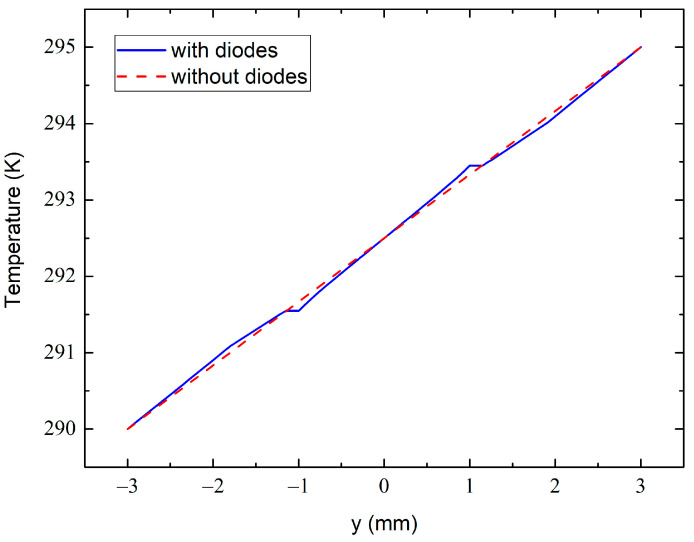
The temperature variation across the dashed line at x= 0.7 mm shown in Figure 11c with (solid line) and without (dashed line) the diode sensors.

**Table 1 molecules-28-07343-t001:** Physical parameters of the constituent phases.

Material	Density, *ρ* (kg m^−3^)	Thermal Conductivity, κ (W m^−1^ K^−1^)	$\mathrm{Kapitza}\;\mathrm{Thermal}\;\mathrm{Boundary}\;\mathrm{Resistance},\;r_{n}$ (W^−1^ m^2^ K)
epoxy	1.200 × 10^3^ [71]	0.22 (our data)	–
graphene	2.267 × 10^3^ [72]	in-plane $\kappa_{g∥}=$ 600–1200 cross-plane $\kappa_{g⊥}=$ = 6 [73]	–
anatase TiO_2_	3.88 × 10^3^ [74]	0.455 [75]	–
graphene–epoxy interphase	–	–	3.5 × 10^−9^ [76]
anatase–epoxy interphase	–	–	(0.1−1.0) × 10^−8^ [77,78]
graphene–anatase interphase	–	–	*

* There are no data available in the literature.

**Table 2 molecules-28-07343-t002:** Parameters of the epoxy/MLG@TiO_2_ composite at 300 K, numerically simulated using Equations (2)–(6).

Interface	$\kappa_{f}$ (W m^−1^ K^−1^)	$R_{K}$ (m^2^·K·W^−1^)	$\kappa_{int}=\frac{h_{int}}{2R_{K}}$ (W m^−1^ K^−1^)	$l_{f}$ (nm)	$\eta_{f}=\kappa_{f}R_{K}$ (nm)	$\kappa_{f1,eff}/\kappa_{f1}$, $\kappa_{f2,eff}/\kappa_{f2}$, $\kappa_{h,eff}/\kappa_{f1}$ (W^−1^ m^2^ K)
Anatase–epoxy	$\kappa_{f2}=$ 0.455	$R_{K}=r_{2}$ = 1.0 × 10^−8^ (1.0 × 10^−9^)	1.2 (12.0)	$l_{f2}=D=$ 50	4.55 (4.55 × 10^−1^)	0.9166 (0.9910)
Graphene (in-plane)–epoxy	$\kappa_{f1}=\kappa_{g11}=\kappa_{g22}=$ 600.0 (1200)	$R_{K}=r_{1}$ = 3.5 × 10^−9^	3.43 (3.43)	$l_{f1}=L_{x}=L_{y}=$ 5 × 10^3^	2.1 × 10^3^ (4.2 × 10^3^)	0.7042 (0.5435)
Graphene (cross-plane)–epoxy	$\kappa_{f1}=\kappa_{g33}=$ 6.0	$R_{K}=r_{1}$ = 3.5 × 10^−9^	3.43	$l_{f1}=L_{z}=$50	2.1 × 10^1^	0.7042
The sandwich-like MLG@TiO_2_ structure						
Graphene (in-plane)–anatase–epoxy	600.0@0.455	$R_{K}=\;r_{2,1}$ = 1.0 × 10^−6^ (1.0 × 10^−7^)		$L_{x}+l_{f2}$	6.0 × 10^5^ (6.0 × 10^4^)	0.00746/0.00752 * (0.0369/0.0384) *
Graphene (cross-plane)–anatase–epoxy	6.0@0.455	$R_{K}=\;r_{2,1}$ = 1.0 × 10^−6^ (1.0 × 10^−7^)		$L_{z}+l_{f2}$	6.0 × 10^3^ ** (6.0 × 10^2^) **	0.01477/0.01489 * (0.0730/0.0760) *
Graphene-wrapped anatase						
Graphene-wrapped anatase–epoxy	606.0@0.455	$R_{K}=\;r_{2,1}$ = 1.0 × 10^−6^ (1.0 × 10^−7^)		${2L}_{z}+l_{f2}$	6.0 × 10^3^ ** (6.0 × 10^2^) **	0.00022 (0.00116)

The values of $\kappa_{int}$ were estimated by inputting $h_{int}=$ 24 nm. * The nominator (denominator) corresponds to $r_{2}=$ 1.0 × 10^−8^ m^2^·K·W^−1^ (1.0 × 10^−9^ m^2^·K·W^−1^). ** In the limit of $l_{f2}⟶\infty$.

**Table 3 molecules-28-07343-t003:** Numerical evaluation of graphene@anatase thermal boundary resistance $r_{21}$.

$\kappa_{C}$ (W m^−1^ K^−1^)	$\kappa_{g11}$ (W m^−1^ K^−1^)	$r_{1}$·10^9^ (m^2^·K^−1^·W^−1^)	$r_{2}$·10^9^ (m^2^·K^−1^·W^−1^)	$r_{21}$·10^7^ (m^2^·K^−1^·W^−1^)
the case of $\kappa_{C}\left(C_{1}=0.01,C_{2}=0.01\right)=\kappa_{C,E1}$ = (0.2864 ± 5%) W m^−1^ K^−1^				
${0.95\kappa }_{C,E1}$	600.0	3.5	1.0 * (10.0) **	0.923 * (0.699) **
$\kappa_{C,E1}$	600.0	3.5	1.0 (10.0)	0.653 (0.429)
${1.05\kappa }_{C,E1}$	600.0	3.5	1.0 (10.0)	0.468 (0.247)
${0.95\kappa }_{C,E1}$	1200.0	3.5	1.0 (10.0)	0.971 (0.749)
$\kappa_{C,E1}$	1200.0	3.5	1.0 (10.0)	0.703 (0.481)
${1.05\kappa }_{C,E1}$	1200.0	3.5	1.0 (10.0)	0.519 (0.297)
the case of $\kappa_{C}\left(C_{1}=0.01,C_{2}=0.05\right)=\kappa_{C,E2}$ = (0.31295 ± 5%) W m^−1^ K^−1^				
${0.95\kappa }_{C,E2}$	600.0	3.5	1.0 (10.0)	5.447 (4.927)
$\kappa_{C,E2}$	600.0	3.5	1.0 (10.0)	4.476 (3.976)
${1.05\kappa }_{C,E2}$	600.0	3.5	1.0 (10.0)	3.745 (3.265)
${0.95\kappa }_{C,E2}$	1200.0	3.5	1.0 (10.0)	5.618 (5.131)
$\kappa_{C,E2}$	1200.0	3.5	1.0 (10.0)	4.644 (4.185)
${1.05\kappa }_{C,E2}$	1200.0	3.5	1.0 (10.0)	3.917 (3.472)

* The values of $r_{21}$ without the brackets correspond to $r_{2}$ without the brackets. ** The values in brackets represent appropriate $r_{2}$ and $r_{21}$ values.

**Table 4 molecules-28-07343-t004:** Volume concentrations of the filler ($\varphi_{1,2,3}$) and interphase layer ($\varphi_{i1,2}$). Top and bottom values correspond to $\rho_{i1}=\rho_{i2}=0.95\rho_{0}$ and $\rho_{i1}=\rho_{i2}=1.05\rho_{0}$, respectively.

$p_{1}$	$p_{2}$	$\varphi_{1}$	$\varphi_{2}$	$\varphi_{3}$	$\varphi_{1}\;+\;\varphi_{2}\;+\;\varphi_{3}$	$\varphi_{i1}$	$\varphi_{i2}$	$\varphi_{i1}\;+\;\varphi_{i2}$
$C_{1}=0.01,C_{2}=0.01$								
0	0	0	0	8.4889 × 10^−3^	8.4889 × 10^−3^	0	1.1266 × 10^−3^	1.1266 × 10^−3^
		0	0	8.4898 × 10^−2^	8.4898 × 10^−2^	0	1.1268 × 10^−3^	1.1268 × 10^−3^
0.05	0.05	2.6799 × 10^−4^	1.5644 × 10^−4^	8.0643 × 10^−3^	8.4877 × 10^−3^	1.1020 × 10^−4^	1.3431 × 10^−3^	1.4533 × 10^−3^
		2.6803 × 10^−4^	1.5647 × 10^−4^	8.0655 × 10^−2^	8.1092 × 10^−2^	1.1022 × 10^−4^	1.3433 × 10^−3^	1.4537 × 10^−3^
0.50	0.50	2.6795 × 10^−3^	1.5642 × 10^−3^	4.2437 × 10^−3^	8.4874 × 10^−3^	1.1019 × 10^−3^	3.2912 × 10^−3^	4.3931 × 10^−3^
		2.6807 × 10^−3^	1.5690 × 10^−3^	4.2456 × 10^−2^	4.6706 × 10^−2^	1.1024 × 10^−3^	3.2927 × 10^−3^	4.3951 × 10^−3^
1.00	1.00	5.3581 × 10^−3^	3.1279 × 10^−3^	0	8.4860 × 10^−3^	2.2034 × 10^−3^	5.4250 × 10^−2^	5.6453 × 10^−2^
		5.3602 × 10^−3^	3.1303 × 10^−3^	0	8.4905 × 10^−3^	2.2051 × 10^−3^	5.4592 × 10^−2^	5.6797 × 10^−2^
$C_{1}=0.01,C_{2}=0.05$								
0	0	0	0	2.1608 × 10^−2^	2.1608 × 10^−2^	0	2.0947 × 10^−3^	2.0947 × 10^−3^
		0	0	2.1612 × 10^−2^	2.1612 × 10^−2^	0	2.0952 × 10^−3^	2.0952 × 10^−3^
0.05	0.05	2.7567 × 10^−4^	8.0463 × 10^−4^	2.0526 × 10^−2^	2.1606 × 10^−2^	1.1336 × 10^−4^	3.3931 × 10^−3^	3.5065 × 10^−3^
		2.7577 × 10^−4^	8.0543 × 10^−4^	2.0533 × 10^−2^	2.1614 × 10^−2^	1.1338 × 10^−4^	3.3943 × 10^−3^	3.5077 × 10^−3^
0.50	0.50	2.7550 × 10^−3^	8.0412 × 10^−3^	1.0796 × 10^−2^	2.1592 × 10^−2^	1.1329 × 10^−3^	1.5071 × 10^−2^	1.6204 × 10^−2^
		2.7594 × 10^−3^	8.0543 × 10^−3^	1.0814 × 10^−2^	2.1628 × 10^−2^	1.1347 × 10^−3^	1.5095 × 10^−2^	1.6230 × 10^−2^
1.00	1.00	5.5060 × 10^−3^	1.6071 × 10^−2^	0	2.1577 × 10^−2^	2.2642 × 10^−3^	2.8028 × 10^−2^	3.0292 × 10^−2^
		5.5227 × 10^−3^	1.6120 × 10^−2^	0	2.1643 × 10^−2^	2.2711 × 10^−3^	2.8113 × 10^−2^	3.0384 × 10^−2^

**Table 5 molecules-28-07343-t005:** Percolation thresholds ($\varphi_{3,PT}$) for hybrid MLG@TiO_2_ nanosheets in MLG@TiO_2_-epoxy nanocomposites in cases of parallelepiped (p)- and spheroid (s)-shaped configurations.

Configuration	$C_{2}$	h (nm)	$\alpha \left(h\right)$	$C_{1,PT}$	$\varphi_{3,PT}$
p-shaped	0.01	4.585	0.0118	0.03121	0.02006
p-shaped	0.05	24.54	0.0196	0.02916	0.03248
s-shaped	0.01	4.571	0.0118	0.03119	0.02005
s-shaped	0.05	24,30	0.0195	0.02890	0.03233

## Data Availability

Data are contained within the article.

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
