# Peer review of "Model Approach to Thermal Conductivity in Hybrid Graphene–Polymer Nanocomposites"

_molecules, 2023, doi:10.3390/molecules28217343_

Round 1
Reviewer 1 Report (Previous Reviewer 1)
Comments and Suggestions for Authors
I thank the authors for addressing my previous concerns by adding COMSOL simulation SEM. A better SEM where the graphite sheet is actually visible is more proper for the final publication otherwise the contrast of the current image or an inset to show the zoomed in region with graphite should be highlighted. The units for the k values for Fig. 10 caption are missing. In fact, Fig 10 can be moved to SI for the coherence of the paper.
Comments on the Quality of English LanguageThe English of the paper is OK
Author Response
Please see attachment

Reviewer 2 Report (New Reviewer)
Comments and Suggestions for Authors
Present article is rather good and clearly written, authors show results of a huge scientific experiment. It is addressed to the synthesis of graphene derivatives, its composite with TiO2 and polymer. Further its characterisation is given together with calculation results. Article can be interesting to the scientists working on material chemistry and physical chemistry, as well as chemical technology. The topic of the research is rather new and relevant to the field of investigation. The possible impact for the scientific area seems average. The references list can be considered as representative. In order to all the conclusions will be supported by the data, authors need to apply HRTEM to characterize carbon and TiO2 materials and composite, since their characteristics are used in calculations and discussion. Unfortunately, Fig.4 is not enough to estimate the size of MLG and TiO2. Authors also need to stress: do they use exfoliated graphite or few-layers graphene. In case of the second choice it would be good to estimate type and amount of functional groups and influence of this interface on thermal characteristics.
Comments on the Quality of English LanguageModerate editing of English language required
Round 2
Reviewer 2 Report (New Reviewer)
Comments and Suggestions for Authors
Accept after minor revision (corrections to minor methodological errors and text editing)
Comments on the Quality of English LanguageMinor editing of English language required
This manuscript is a resubmission of an earlier submission. The following is a list of the peer review reports and author responses from that submission.
Round 1
Reviewer 1 Report
Comments and Suggestions for Authors
This paper discusses the properties of polymer nanocomposites based on graphene and its derivatives. The authors explore the structural, mechanical, electrical, and thermal properties of these nanocomposites. The authors focus on understanding the impact of filler content on composite properties and the interphase interaction between the polymer and graphene. While the paper provides valuable insights into the understanding of a specific nanocomposite, one potential limitation is the difficulty in quantitatively analyzing the thermal conductivity of the nanocomposites. This manuscript should be published after major revision by considering my following questions and comments.
1. The thermal conductivity measurement is not reliable enough in multiple aspects. Why do the authors choose to use electrical diodes with dimension around 1/10 of the size of the sample which is apparently not consistent with the ASTM guarded hot plate method they cited? The plastic cover of this NXP BAP64-02 can add to the thermal resistance between the sensor and the sample. The thermal mass and low thermal conductivity of this sensor is too big to be ignored which can significantly interrupt the temperature field of the measurement. In general heat conduction experiment, micron level metallic thermocouple or RTD is more commonly used to minimize the sensor influence. In addition, it is unclear how the authors insulated the side walls of the sample to reduce the error from the edge loss, and how they quantify the gap loss, radiation loss of the rough composite and flux imbalance between the sample and the heater and other factors.
2. In fig 3, there’s no Si, which is inconsistent with line 242. Also, why don’t the author use experimentally measured SiO2, i.e., quartz, thermal conductivity vs. temperature?
3. The authors need to at least show some SEM or other microscopic images to demonstrate the internal structure of the composite. This is needed to justify the so called SLG model they used later which can be completely invalid if the TiO2 does not touch the graphene.
4. The interface thermal resistance of the graphene-epoxy interface is from Ref. 82 in the early Raman experiment is too small compared with many later measured as listed in ACS Appl. Mater. Interfaces 2016, 8, 12, 8272–8279 and International Communications in Heat and Mass Transfer, 131, 2022, 105846 This can significantly affect the results in this paper.
5. Why is the R_k for the in-plane graphene-epoxy interface the same as that for the cross plane? This should be a complete different thermal contact in which the graphite phonon DOS is shifted to high energy due to the in-plane vibration compared to the cross-plane case. The sound velocity is also significantly different. In terms of the impact of the nanoscale thickness and interface impact of the thermal conductivity of the MLG, this work should be considered and cited: Advanced Materials Interfaces 3 (16), 1600234.
6. What value did the authors use in Table 2 for h_int and why?
7. For the thermal conductivity of MLG nanosheet and the anatase particles, how is the thickness and diameters measured?
8. The parameters of k_f and k_m are not defined before they first appear.
9. There is a mixed use of kappa and k to represent thermal conductivity in the equations.
10. What is the physical meaning of ?? in Eq. 9?
11. Why do the authors use r2 = 1e-9 m2·K·W-1 which is out of the range of 0.3e-8-1e-8 m2·K·W-1 given in Table 1? This makes no sense.
12. In the model used Fig. 5-9, many parameters are too arbitrary, such as the interface layer thickness, the in-plane thermal resistance, and the filler geometry. Why do they use 10 nm for h_int when the MD simulation yields 12-14 nm? On the other hand, the loading volume and mass fractions should be known but its is not clearly stated how these are measured.
13. I don’t see direct evidence of the so-called self assembly in the composite.
14. Importantly, in Fig. 5 and below, the authors need to provide a clear detailed comparison of the model prediction and experimental results.
15. Eq1 is not meaningful as the authors didn’t even use the specularity and size to estimate the thermal conductivity of the nanofillers.
16. The sentence in line 46-48 is too long and appears incomplete.
In terms of impact, the paper contributes to the existing body of knowledge on polymer nanocomposites, but it may not be considered groundbreaking due to its focus on a specific aspect of the topic and the limitations mentioned. Nonetheless, it somewhat consolidates the understanding of interphase interactions and highlights the need for further research in this area.
Comments on the Quality of English Language
Overall the English is decent. Some minor issues in the equations is mentioned above. This paper need a list for the variable in the appendix since it contains various parameters in the model.
Reviewer 2 Report
Comments and Suggestions for Authors
This work is concerned with the effective thermal conductivity of self-assembled hybrid graphene/anatase-epoxy nanocomposites. The study combined theoretical modeling and experimental measurements over the temperature range from 40 K to 320 K. In the theoretical part, the Kapitza interfacial resistance was analyzed following the effective-medium approach of Su et al. (Carbon, 2018), which was extended to cover the hybrid graphene@anatase nanosheets either of parallelepiped- or spheroidal-like configurations. Temperature-dependence of thermal conductivity for neat epoxy and the hybrid nanocomposites are presented and analyzed. The Introduction is insightful, and Results and Discussion are rich. It is highly recommended for publication in Molecules.
There are two typos:
(i) In the Reference list, Ref. [12] and Ref. [98] appear to be identical.
(ii) in Ref. [105], the name of the first author should be Wang, Y., instead of Yang, W., that is, Wang, Y., Shan, J.W., Weng, G.J.
In summary, the subject of hybrid nanocomposites is receiving increasingly wide attention. This paper will make a timely contribution to literature.
Reviewer 3 Report
Comments and Suggestions for Authors
The manuscript by Nadtochiy and coworkers addresses the combined experimental and theoretical study of the thermal conductivity, k, of epoxy nanocomposites enriched with hybrid graphene-TiO2 nanoparticles, focusing of the temperature dependence of k.
The work is well contextualized and quite well organized.
The investigation architecture is clear and is based on the synergic combination of experimental and modeling efforts.
The thermal conductivity response vs T is reported in a wide T range (40-350 K) for different families of epoxy-based samples, including different compositions and structures.
The key parameters of both starting phases and final samples are reported.
On the modeling side, the authors addressed different architectures of the nanocomposites, calculate related thermal boundary resistance, volume concentrations of the filler and interphase layer, and discussed the loading dependencies of k.
The key material properties are finally parametrized (e.g. reporting k and normalized k vs thermal boundary resistance
Perspectives for chemical engineering strategies towards a controlled k in the investigate samples are discussed.
The work seems solid and rigorous and overall adds some novelty to the field
No weakness are evident .
Author Response
Thank you very much for your kind support
Round 2
Reviewer 1 Report
Comments and Suggestions for Authors
I thank the authors for answering my question. However, multiple serious concerns remain.
1. Where does the dimensions the particle and nanoplate such as the ones used in Eq. 10 and Table 2 come from? The STM images they used do not show any identifiable particle to allow a statistical analysis of the TiO2 and graphite sizes. Instead of STM, the authors need to use TEM to directly see the particles by make a cross section slice to justify all the analysis. Otherwise, there's absolutely no reason to use the arbitrary numbers of "?? = 50 nm, ?? = ?? = 5x103 nm" in Eq. 10 to claim that the in and cross plane thermal resistance is similar. For this kind of thermal model of composites, authors should either use directly observed geometry from SEM, TEM or treat the dimensions as a fitting parameter and use sensitivity analysis to justify the fitting reliability.
2. Indeed they added the value for h_int in Table 2,. but similar to the case in 1, why do they choose to use 10 nm is still missing. Where does the 5 nm of R_G come from? This is again just pure guess without using scaling law.
3. The revised explanation for the use of the diode is still unsatisfactory. The method used by the authors is a steady state guarded hot plate method which assumes the heat transfer is 1D along the vertical direction. The issue is that even the smallest dimension of the sensor, 0.6 mm is too large compared with the total thickness of the sample along the vertical direction, 6 mm. The two sensors cover 1/5 of the height of sample in fact. It means that the temperature measured is not representing it for an exact point but an average of a tall region along the temperature profile. Even if the temperature profile is linear as the authors assumed, the experimental data will become an average of the multiple metal leads. I also don't understand what the author meant by transient measurement in line 237.
4. The data of which sample from the experiments in Fig. 3 is used for comparison in Fig. 8,9? Apparently the concentration should be different for different curves in Fig. 8,9 but the authors seem to be using a single experimental k value which inconsistent with Fig. 3. The r12 values obtained from such comparisons with experiment appear to vary by orders of magnitude which makes the whole model unreliable.
5. The objective of this work is to show the effect of a newer thermal model for nanocomposite. But the current sample is too complicated (involves too many parameters) to allow one to unambiguously resolve useful TBR. Can the authors try the model on pure TiO2 epoxy which should have much less geometry parameters to show its reliability?
Comments on the Quality of English LanguageThe English is OK but a the readability is still low. For example in line 228 the lead thermal resistance of the lead is given which does not include the lead/epoxy contact. What should we compare it to? Is this value small? What is the meaning of m_g in Eq1? I highly recommend the authors to prepare a list to show the meaning of each parameter they used in the model and how they are related.
